# Stochastic alluvial fan and terrace formation triggered by a high-magnitude Holocene landslide in the Klados Gorge, Crete

Elena T. Bruni[1], Richard F. Ott[1], Vincenzo Picotti[1], Negar Haghipour[1, 2], Karl W. Wegmann[3, 4], Sean F. Gallen[5]*

[1] Department of Earth Sciences, ETH Zürich, Switzerland
[2] Laboratory of Ion Beam Physics, ETH Zürich, 8092 Zurich, Switzerland
[3] Department of Marine, Earth and Atmospheric Sciences, North Carolina State University, Raleigh, NC, USA
[4] Centre for Geospatial Analytics, North Carolina State University, Raleigh, NC, USA
[5] Department of Geosciences, Colorado State University, Colorado, USA

*Correspondence to*:      Sean F. Gallen (sean.gallen@colostate.edu)
                          Elena T. Bruni (elena.bruni@erdw.ethz.ch

**Abstract**
Alluvial fan and terrace formation are traditionally interpreted as a fluvial system response to Quaternary climate
oscillations under the backdrop of slow and steady tectonic activity. However, several recent studies challenge
this conventional wisdom, showing that such landforms can evolve rapidly as a geomorphic system responds to
catastrophic and stochastic events, like large magnitude mass-wasting. Here, we contribute to this topic through
a detailed field, geochronological and numerical modelling investigation of thick (> 50 m) alluvial sequences in
the Klados catchment in southwestern Crete, Greece. The Klados River catchment lies in a Mediterranean climate,
is largely floored by carbonate bedrock, and is characterised by well-preserved, alluvial terraces and inset fans at
the river mouth that exceed the volumes of alluvial deposits in neighbouring catchments of similar size. Previous
studies interpreted the genesis and evolution of these deposits to result from a combination of Pleistocene sea-
level variation and the region's long-term tectonic activity. We show that the > 20 m thick lower fan unit,
previously thought to be late Pleistocene in age, unconformably buries a paleoshoreline uplifted in the first
centuries AD, placing the depositional age of this unit firmly into the Late Holocene. The depositional timing is
supported by seven new radiocarbon dates that indicate mid to late Holocene ages for the entire fan and terrace
sequence. Furthermore, we report new evidence of a previously unidentified valley-filling landslide deposit that
is locally 100 m above the modern stream elevation, and based on cross-cutting relationships, pre-dates the alluvial
sequence. Observations indicate the highly-erodible landslide deposit as the source of the alluvial fill sediment.
We identify the likely landslide detachment area as a large rockfall scar at the steepened head of the catchment.
A landslide volume of $9.08 \times 10^7 \, \text{m}^3$ is estimated based on volume reconstructions of the mapped landslide deposit
and the inferred scar location. We utilise landslide runout modelling to validate the hypothesis that a high
magnitude rockfall would pulverize and send material downstream, filling the valley up to ~ 100 m. This partial
liquefaction is required for the rockfall to form a landslide body of the extent observed in the valley and is
consistent with the sedimentological characteristics of the landslide deposit. Based on the new age control, and
the identification of the landslide deposit, we hypothesise that the rapid post-landslide aggradation and incision
cycles of the alluvial deposits are not linked to long-term tectonic uplift or climate variations but rather stochastic
events such as mobilisation of sediment in large earthquakes, storm events, or ephemeral blockage in the valley's
narrow reaches. The Klados case study represents a model-environment for how stochastically-driven events can
mimic climate-induced sedimentary archives, lead to deposition of thick alluvial sequences within hundreds to
thousands of years, and illustrates the ultrasensitivity of mountainous catchments to external perturbations after
catastrophic events.

## 1 Introduction

Alluvial fans and terraces are traditionally used as proxies for climate variations and tectonic activity. Within this view, their formation depends on climate-driven changes in the ratio of sediment supply and transport capacity superimposed on the long-term tectonic activity of the region (Bridgland et al., 2004; Bull, 1991; Merritts et al., 1994; Pazzaglia, 2013; Schumm, 1973). However, an increasing number of studies report that stochastic mechanisms such as landslides and autogenic fluctuation in river channel positions can also generate these landforms (Finnegan et al., 2014; Korup et al., 2006; Limaye and Lamb, 2016; Scherler et al., 2016). Such stochastically-generated deposits can resemble climate-forced alluvial terraces and fans in structure and sedimentology, possibly leading to erroneous interpretations of the processes responsible for their genesis. However, it is possible to distinguish between climatic and stochastic mechanisms for fluvial terrace and fan formation through careful field observation, precise geochronology, and comparisons to regional climate records (c.f., Scherler et al., 2016). Nevertheless, it remains unclear how rivers and river catchment systems react and recover from the high-magnitude stochastic perturbations (i.e., a large landslide) that can rapidly build thick alluvial sequences. However, modelling studies may provide a basis for interpretation(i.e., Hungr and Evans, 2004). Furthermore, little is known about how such catastrophic events alter earth surface dynamics, which might generate different responses to superimposed variations in climate and tectonics in affected and unaffected catchments. Here we contribute to the growing body of literature on the role of climatic versus stochastic mechanisms as a driver of rapid emplacement of fluvial landforms and the impacts of stochastic forcing on catchment-scale earth surface dynamics through the investigation of an exemplary fill sequence of thick (> 50 m) alluvial fan delta and terrace deposits, in a small, steep, mountainous catchment on the southern coast of Crete, Greece (Fig. 1a).

Previous studies on Crete and elsewhere in the Mediterranean show that the construction of Quaternary alluvial terraces and fans is generally linked with climate fluctuations (Gallen et al., 2014; Macklin et al., 2010; Nemec and Postma, 1993; Pope et al., 2008; Wegmann, 2008). Also, human land use and vegetation cover have been shown to influence sediment dynamics and alluviation patterns, and the Eastern Mediterranean has been central to the investigation of the interplay between climate fluctuations, long-term tectonics, and anthropogenic disturbances (Atherden and Hall, 1999; Benito et al., 2015; Dusar et al., 2011; Thorndycraft and Benito, 2006; Vita-Finzi, 1969). A key study site is the large Bajada-type alluvial fan system on the Sfakian piedmont of southern Crete ~30 km east of the Klados catchment. The Sfakia fan sequence was initially mapped and described by Nemec and Postma (1993) with subsequent detailed chronology developed by Pope et al. (2008), Ferrier and Pope (2012), and Pope et al. (2016) using luminescence dating and soil chronostratigraphy. From sedimentology, topographic surveys, soil redness indices, and chronometric dating, the Sfakian fans are interpreted as recording sediment deposition during colder and wetter glacial stages with little to no fan deposition during the intervening warm interglacial or interstadial periods (Pope et al., 2008, 2016). This result agrees with Gallen et al. (2014), who found that alluvial fans on the south-central coastline of Crete aggraded and prograded in response to increased catchment delivery of sediment and the lowering of the sea level (base level) during cold climate intervals. Similar conclusions were drawn by Wegmann (2008) and Macklin (2010), who found that active fan aggradation on Crete generally occurred during glacial stages. These examples illustrate that fan sedimentation occurred more or less in concert with Quaternary climate variability across western and southern Crete, similar to

elsewhere in the Mediterranean (Benito et al., 2015; Macklin and Woodward, 2009; Thorndycraft and Benito, 2006; Zielhofer et al., 2008).

Most Cretan alluvial deposits share commonalities in stratigraphy, sedimentology, pedogenesis, and aggradational chronology. However, the thick sequence of several > 20 m thick alluvial fan and terrace deposits preserved in the Klados catchment is anomalous compared to nearby catchments with larger drainage areas (i.e., Samaria) that preserve only minor alluvial deposits. The Klados River catchment drains the south flank of Volakis Mountain (2,116 m), which features a steep, 42° planar slope that dips southward off the mountain's upper flank (Fig. 1b), and is surrounded by steep, 2 km high mountains, which has kept human influence minimal. The stream incises metamorphosed Jurassic to Eocene Plattenkalk limestone (Creutzburg, 1977). Two large inset fans are present at the catchment mouth; they extend ~650 m along the beach between adjacent bedrock promontories (Fig. 1e). Wave action eroded the fan deposit toes, forming sea cliffs up to 30 m high (Fig. 1e). Each fan grades upstream into thick (> 20 m), well-preserved paired (valley-spanning) fill terraces. Consequently, Klados fan terrace deposit volumes are oversized relative to the relatively small catchment area (11.5 km$^2$), which is particularly evident when compared to deposits in the adjacent gorges with larger drainage areas. Furthermore, the alluvial fan-terrace deposits display little weathering and immature soil development, especially relative to the well-studied Late Pleistocene alluvial fan sequences preserved along the south coast of Crete (Gallen et al., 2014; Macklin et al., 2010; Nemec and Postma, 1993; Pope et al., 2008; Wegmann, 2008).

Previous research on the Klados alluvial sequence focused on the deposits in proximity to the sea at the stream mouth. One study argues that these deposits aggraded in the Holocene due to short-term climate fluctuations, rapid uplift rate, and variations in sediment supply (Booth, 2010). In contrast, another study suggests that the fans are associated with Late Pleistocene climate-eustatic fluctuations and long-term tectonic uplift based on field observations and luminescence dating (Mouslopoulou et al., 2017). The correct interpretation of these exceptional deposits has important implications for understanding the role of climatic versus stochastic mechanisms on catchment-scale sediment transport, alluvial fan and terrace development, and the use of alluvial deposits as environmental archives in Crete and elsewhere.

Due to the unique appearance of the Klados sedimentary deposits, conflicting interpretations of their timing and genesis, and ambiguous geochronology, this study revisits the origin and evolution of alluvial deposits within and at the mouth of the Klados Gorge. The volumes of these deposits are substantially larger than alluvial deposits in larger neighbouring catchments and therefore require an unusually high sediment input. Through mapping and cross-cutting relationships, we show that a previously unidentified valley-filling landslide deposit - locally more than 100 m thick and in many locations covering paleo-bedrock topography - is the source of sediment feeding the Klados alluvial fan delta. Relative and absolute geochronology places the landslide deposit and the construction of the alluvial fans and terraces firmly in the Holocene, contrary to previous luminescence ages. Our mapping, coupled with landslide runout modelling, suggests that a high-magnitude, catastrophic mass-wasting event in the catchment headwaters backfilled the valley. The rapid input of large quantities of sediment into the catchment provides an excellent opportunity to investigate how rivers respond to such catastrophic events.

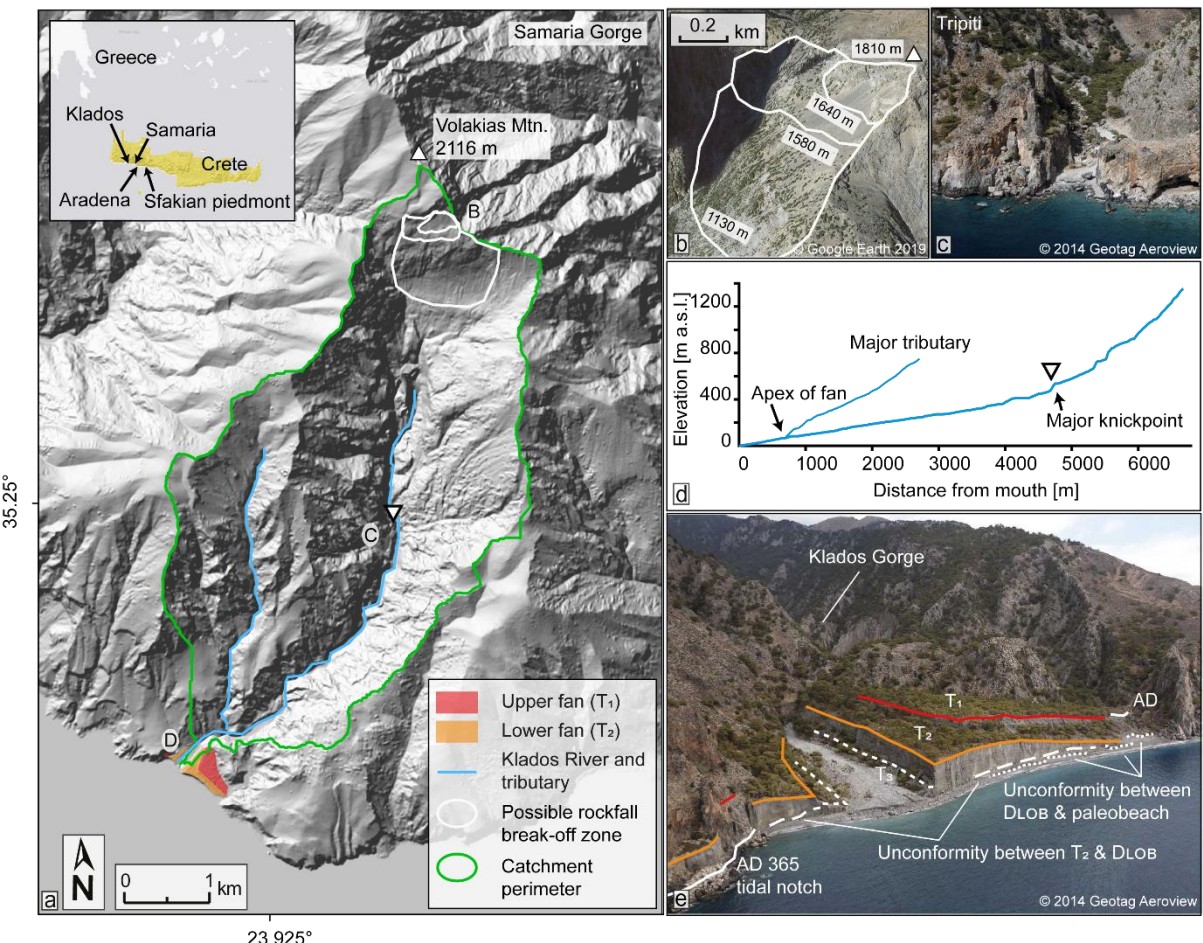

**Figure 1**: Overview of the Klados catchment and fan. (a) Hillshade of the Klados river catchment with the alluvial fan delta at its mouth (coloured). Note the steep planar surface at the head of the catchment, which we interpret as a rockfall failure plane. The extents of the minimum, intermediate, and maximum rockfall areas are outlined in white as used in the landslide modelling (Sect. 5.3). Inset overview of Klados catchment location on Crete, Greece, with the study area and other relevant locations indicated (ESRI, 2011). The hillshade was generated from the 5 m DEM of the Hellenic Cadastre SA. (b) Oblique perspective Google Earth view of the hypothesised failure plane(s) at the head of the catchment (outlined as in a). (c) The Tripiti catchment outlet < 5 km to the west of the Klados fan sequence. Note the absence of large alluvial deposits, typical for the rivers draining the Levka Ori Moutains, highlighting the uniqueness of the deposits at Klados. (d) River longitudinal profiles of the Klados River and its major tributary. (e) Oblique aerial photograph of the alluvial fans delta at the Klados catchment outlet. Highlighted are the different surfaces and unconformities discussed in the text.

## 2 Regional setting

Crete is in the forearc of the Hellenic subduction zone, where the African plate subducts beneath the Aegean microplate at a rate of ~35 mm $a^{-1}$ (McClusky et al., 2000; Reilinger et al., 2006). The crust beneath Crete consists of a compressional nappe pile built during subduction in the mid-Cenozoic and exhumed in the Late Cenozoic (Fassoulas et al., 1994; van Hinsbergen and Meulenkamp, 2006). Miocene to Pliocene marine sediments in-filled extensional basins. These basins subsequently uplifted several 100s of meters and are now exposed above sea level on Crete (van Hinsbergen and Meulenkamp, 2006; Meulenkamp et al., 1994; Zachariasse et al., 2011). Quaternary paleoshorelines document ongoing uplift; some are now hundreds of meters above sea level (a.s.l.) (Angelier et al., 1982; Gallen et al., 2014; Ott et al., 2019b; Robertson et al., 2019). Craggy cliffs interrupted by deeply-incised valleys and bedrock gorges characterise southwest Crete's coastal topography, where basin-average erosion rates are ~ 0.1 mm/a (Ott et al., 2019a).

The island lies above the most active seismic zone in the Mediterranean, and episodic Holocene uplift in western Crete associated with earthquakes occurs under the backdrop of slower steady rock uplift driven by deeper crustal processes (Gallen et al., 2014; Ott et al., 2019b; Pirazzoli et al., 1982; Shaw et al., 2008; Stiros, 2001). Evidence of large earthquakes comes from historical reports, archaeological excavations, tsunami deposits, and uplifted Holocene paleoshorelines (Ambraseys, 2009; Dominey-Howes et al., 1999; Pirazzoli et al., 1996; Shaw et al., 2008). These paleoshorelines delineate the temporal position of sea level through tidal or bioerosional notches, cemented beachrock, topographic benches, and shore platforms (Chappell, 2009). The uplift of a Holocene paleoshoreline by as much as 9 m a.s.l. on the southwestern coast of Crete is often attributed to an unusually large earthquake ($M_W$ 8.3–8.5) in AD 365 (Mouslopoulou et al., 2015; Shaw et al., 2008), but a more recent study suggests that uplift occurred through a series of earthquakes with $M_w < 7.9$ in the first centuries AD (Ott et al., 2021). Regardless of conflicting interpretations, this prominent paleoshoreline is observable along > 200 km of coastline in western Crete and provides a robust Late Holocene time marker. Following Ott et al. (2021), we refer to this Late Holocene coastal feature as the Krios paleoshoreline, based on its maximum elevation at Cape Krios in southwestern Crete. At the mouth of the Klados catchment, the Krios paleoshoreline is preserved as a tidal notch and beach deposit at 6 m a.s.l. We use contact relationships between the Krios paleoshoreline and alluvial deposits as a relative age marker.

## 3 Field and laboratory methods

### 3.1 Field observations and spatial analysis

Field mapping was complemented by spatial analysis of digital elevation models (DEM) using ArcGIS v10.2 and TopoToolbox v2 (ESRI, 2011; Schwanghart and Scherler, 2014). The field mapping focused on stratigraphic and sedimentological characteristics of the Late Quaternary deposits, cross-cutting relationships, and the degree of soil formation that allowed for the classification of distinct geomorphic units (IUSS Working Group WRB, 2015). A 5 m DEM of the catchment, provided by the Hellenic Cadastre SA, was used to determine the longitudinal river profile and the heights of the terraces above the modern channel elevation. The DEM was also used to reconstruct the extent and volume of eroded Quaternary alluvial fill deposits. While a higher-resolution DEM was produced by a photogrammetric analysis of drone imagery (AgiSoft, 15 cm resolution) and was subsequently used for the diffusion modelling (Supplement sect. 7; Fig. S6), we did not use it for volume estimations because of vegetation coverage. We used elevation data from mapped geomorphic units that exhibited little erosion to model a given deposit's pre-incision surface for the volume reconstruction. These pre-incision surfaces were constructed via spline (regularised, weight = 0.1) interpolation in ArcGIS. To determine eroded volumes and the original extent of the deposits, we subtracted the modern DEM from the interpolated pre-incision surfaces. Subsequently, a reconstruction of the pre-deposition valley bedrock morphology was created by subtracting the mapped thicknesses of individual deposits from the modern DEM.

### 3.2 Radiocarbon dating

#### 3.2.1 Fossil dating

Two in-situ Vermetid (sessile marine gastropod) shells were sampled from the uplifted Late Holocene Krios paleoshoreline at 5 and 6 m a.s.l. These data were used to locally constrain the age of the uplifted Krios paleoshoreline, presumed to be upheaved in the first centuries AD, and provide a maximum depositional age for the younger alluvial fan delta. The samples were crushed, washed in 0.06% HCl, and infused with 85% phosphoric acid. After graphitisation of the released $CO_2$ in an AGE3 system (Wacker et al., 2010), the resulting ~ 1 mg of graphite was analysed by an Accelerator Mass Spectrometer (AMS). The standards used in the graphitisation step are 8.55–9.12 mg IAEA-C1 carbonate and 9.97–10.54 mg IAEA-C2 I Travertine carbonate. We also collected a terrestrial gastropod shell from the upper fan surface in reddish silt to constrain the minimum age of fan surface abandonment. The radiocarbon ages are reported in fraction modern (fm) values and radiocarbon years (yr) with a 1 σ range. The fossil radiocarbon ages were calibrated using OxCal (Bronk Ramsey, 2009) to compare the inferred Krios paleoshoreline emergence in the first centuries AD. The Marine13 calibration curve was used for calibration (Reimer et al., 2013) with a marine reservoir effect of 58 +/- 85 years as suggested by Reimer and Reimer (2001). We report the 2-σ ranges of calibrated years before the present (1950 AD, calBP.).

#### 3.2.2 Bulk sediment dating

To constrain the timing of aggradation and incision of the deposits, we radiocarbon-dated bulk organic matter collected from six fine-grained lenses within the deposits. While bulk radiocarbon dating of alluvial sediments will result in larger uncertainties, in this case, it is the only available geochronometric technique given the mineralogy of the sediments and lack of macro-organic material for traditional AMS radiocarbon dating.

Additionally, despite uncertainties associated with bulk radiocarbon dating, it is appropriate for discriminating
whether or not the sediments are late Pleistocene or Holocene, one of the hypotheses tested with this study. We
decided against using luminescence dating because of the sparsity of quartz and feldspar in the local carbonate
bedrock and the turbulent mode, and the short transport distance likely resulting in incomplete bleaching,
especially of feldspar grains (Rhodes, 2011). A detailed discussion of uncertainties associated with this method is
provided in sect. 5.1.

The samples consist of 0.02 to 0.03 wt. % of total organic carbon. These samples were extracted, fumigated with
HCl at 70 °C for three days, and neutralised using NaOH (McIntyre et al., 2017). Due to low TOC, we measured
two gas target runs with ~ 80 mg of sample in the first and ~ 120 mg in the second. The samples have been
corrected for constant contamination correction using shale (fm=0.018 and Swiss soil fm=1.06) with a Matlab
code described in Haghipour et al. (2019). The radiocarbon ages are reported in fraction modern (fm) values and
years (yrs.) with a 1-σ range.
**3.3  Parameters used in the landslide modelling**
To test the feasibility of the hypothesis that a rockfall turned landslide provided the necessary material to form
the large sedimentary deposits throughout the valley, we utilised the DAN3D-Flex dynamic landslide runout
model that allows an initial coherent phase of motion followed by the flow-like movement of the rock mass (Aaron
et al., 2017). Several studies report successful model results for landslides when a Voellmy or frictional rheology
is used as the basal rheology, and several back-analysed historical events are available using these rheologies
(Aaron and Hungr, 2016; Grämiger et al., 2016; Hungr, 1995; Nagelisen et al., 2015). Voellmy rheology adds a
"turbulent term" to the basic frictional rheology equation, which is dependent on flow velocity and the density of
the material and summarises the velocity-dependent factors of flow resistance (Hungr and Evans, 1996).
Moreover, the model requires input files containing the pre-failure surface combined with the topography of the
sliding surface over which the slide flows ("path topography") and the vertical depth of the sliding mass at the
initial position represented by the source material isolated from its surrounding ("source depth").

Input parameters of topography, sliding surface, and volume were estimated and calculated based on the modern
topography. We produced a DEM of the modern landscape without the Holocene deposits mapped in this study
as the pre-landslide topography (DEMpre). For this, the thicknesses of all deposits were subtracted from the
present-day topography (Fig. S2). The pre-failure surface for the source area was reconstructed using the
thicknesses of the reconstructed rockfall wedges creating a rough minimum estimate of the mountain face's
bedrock topography before the landslide event. The thicknesses were assumed to correspond to the elevations of
the terraces from the modern river bed. DEMpre was also used to estimate the volume of the initial landslide
valley infill. We interpolated a horizontal plane of constant elevation at the maximal elevation of the landslide
deposit at 100 m and subsequently measured the vertical distance from this plane to the pre-landslide topography
(DEMpre). This calculation provides an estimate of the volume necessary to reach the given plane of elevation,
even though it neglects the effects of topographic obstruction by the narrow valley reaches. Additionally, we
approximated the possible maximum and minimum volumes for the valley infill by varying the landslide deposit
elevation by 20 m.
We calculated several scenarios for the initial amount of material detached from the mountain face and compared
them to the volumes of the valley infills produced as described above. For the calculations in ArcGIS, we
constructed a point cloud on the source area which was then transferred to a multipatch feature that fully encloses
the input data and thus, visualises the volume of the missing material (Fig. S3). The thickness of the pre-failure
rock slab on the mountain-face scarp was estimated based on a critical friction angle (30°) and the extent of the
modern planar surfaces (Figs. 1b; S3). The best-fitting volumes between the valley infills and the wedges were
subsequently used to approximate the rock fall's initial size and model the landslide runout.

## 4 Results

The Klados catchment contains three generations of alluvial infill units (denoted as $T_1$, $T_2$, and $T_3$, from highest to lowest, respectively) extending from the river headwaters to the beach, where they terminate in large telescopic fans (Figs. 2; 3a). The two upper alluvial infills ($T_1$ and $T_2$) form steep coastal cliffs separated from the sea by a 2-10 m wide cobble-pebble beach while the lower fill ($T_3$) grades to more recent fluvial gravels downstream. The Klados River incised into each of the alluvial deposits, forming terrace treads at ~ 50, 20, and 5 m above the modern channel, respectively. An additional deposit ($L_1$) is found as high as 100 m above the channel and has an irregular basal contact that fills in a paleo-bedrock topography. It does not grade into a fan, and its sedimentology is distinct from the alluvial deposits, suggesting formation by a different process.

Deposits $T_1$ and $T_2$ cut and form buttress unconformities in various locations against the $L_1$ deposit (Fig. 4a), indicating that the alluvial deposits were emplaced after the deposition of $L_1$. Basal unconformities (i.e., straths) of the alluvial and $L_1$ deposits rarely crop out in the lower reaches but are increasingly visible upstream (Fig. 3a). Nevertheless, we identified that $T_2$ unconformably overlies a paleo-beach deposit (Fig. 4b). $T_2$ gravels also cover Vermetid shells growing in the tidal notch (i.e., the Krios paleoshoreline) that lies at the same elevation as the paleobeach deposit demonstrating that the $T_2$ unit post-dates the Late Holocene paleoshoreline features (Fig. 5c). An eolian deposit locally tops the $T_1$ surface proximal to the seaward cliff. In addition to these prominent deposits, we identified several smaller, more recent infills distributed over the catchment's lower reaches.

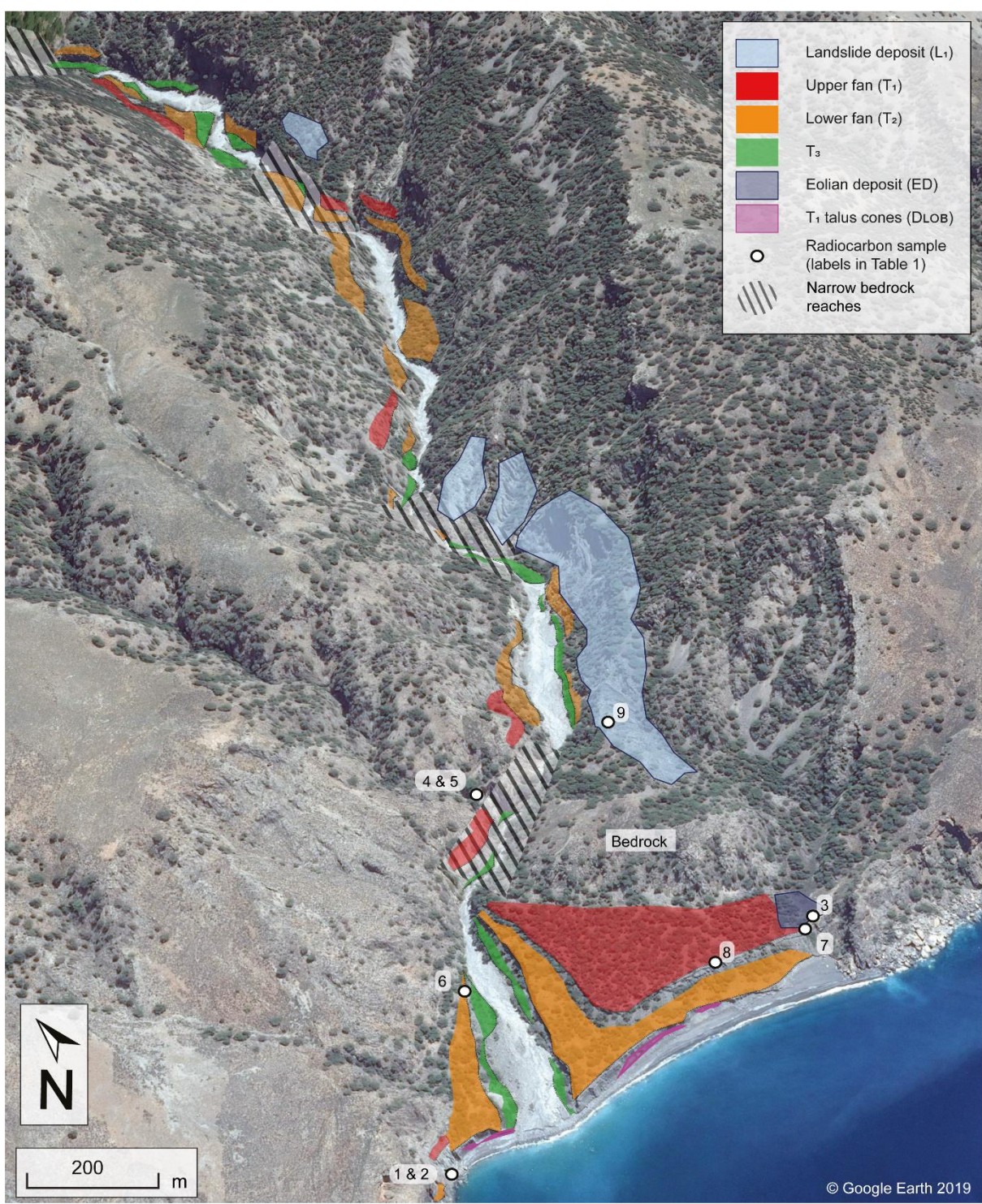

264

Figure 2: Overview of Quaternary deposits in the lower half of the Klados catchment. The numbers indicate the location of radiocarbon samples (labels correspond to Table 1). Note the inset fan surfaces at the outlet of the modern river mouth.

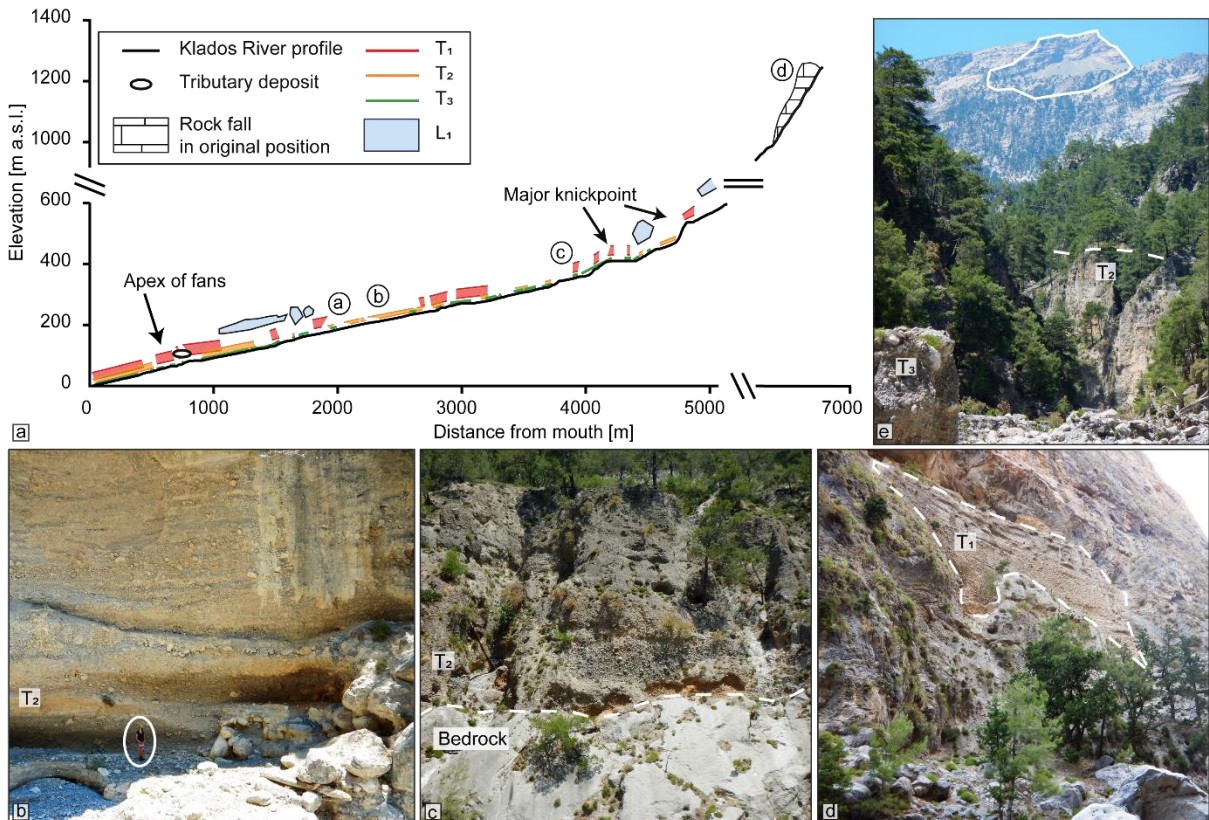

Figure 3: (a) Klados River longitudinal profile. Both the top tread and the bottom contacts of the mapped infills can be traced from the headwaters to the sea, where they form the massive alluvial fans. The hypothesised position of the initial rockfall is indicated. (b) The valley is filled with thick (> 30 m) terraces (person for scale). (c) Contact between bedrock and $T_2$ valley infill. (d) The $T_1$ valley infill deposits are found up to 60 m above the modern river channel. (e) Distant view of the potential rockfall source on Volakias Mtn. (highlighted).

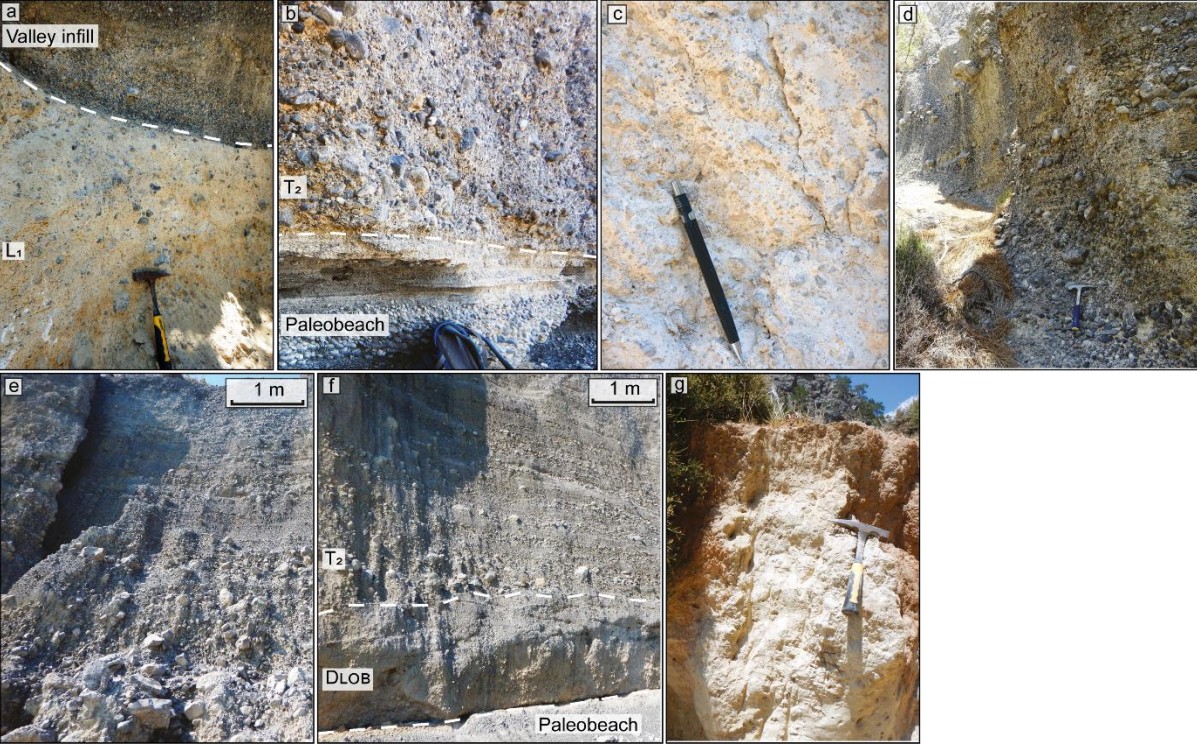

**Figure 4**: Images of the main deposits and their unconformities. (a) Unconformity between the landslide deposit ($L_1$) and one
of the valley infills. (b) Paleobeach buried by $D_{LOB}$ and $T_2$ close to the modern river channel. (c) Typical appearance of the
landslide deposit characterised by angular clasts floating in a fine matrix. (d) The top section of the upper fan ($T_1$) is
characterised by cobble to pebble-dominated planar beds typical of flow in shallow channels. (e) $T_2$. Note upward grading
from unsorted, subangular boulders and cobbles to a laminar deposit similar to (d). (f) Paleobeach buried by $D_{LOB}$ and $T_2$ close
to the modern river channel. (g) Eolian deposit from the eastern fan ($T_1$) surface (ED).

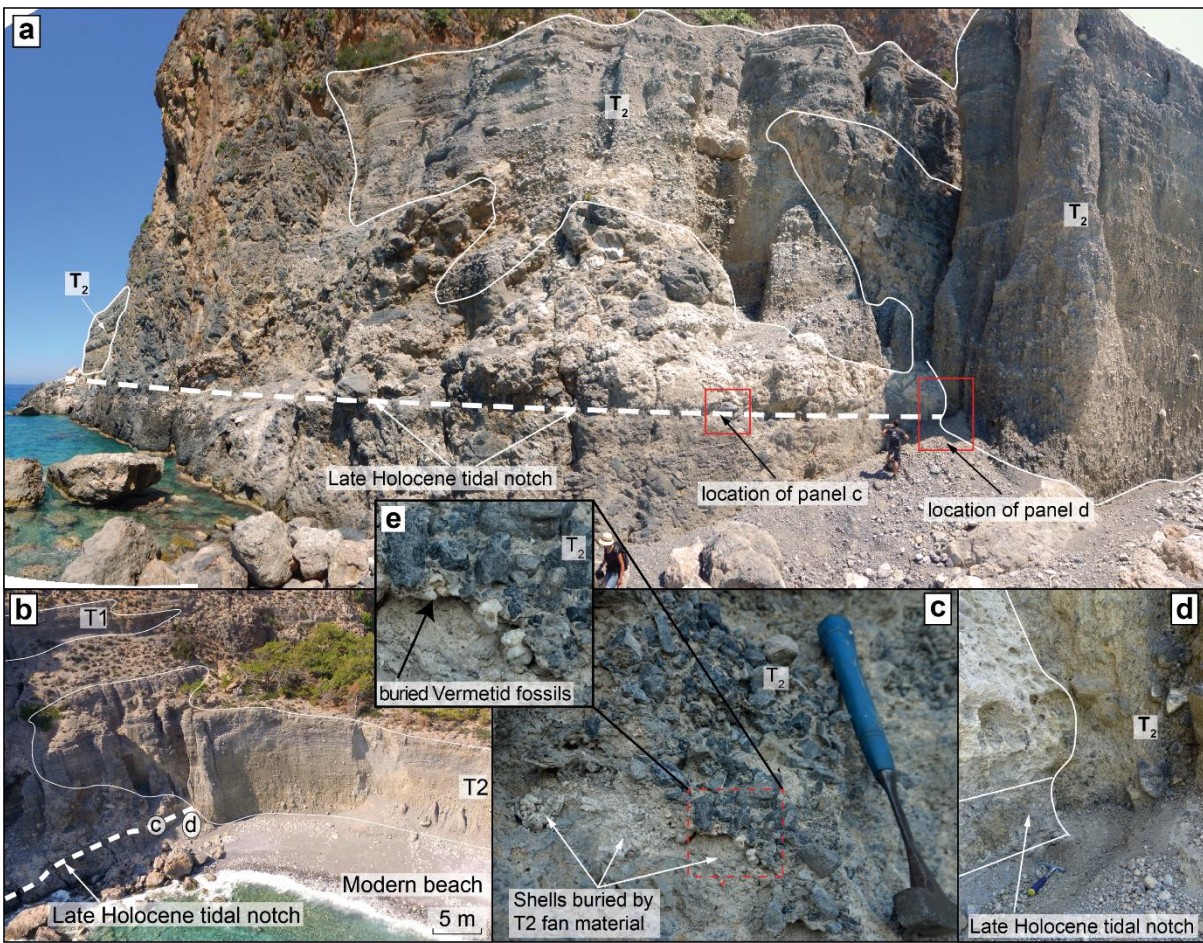


**Figure 5**: The contacts between the tidal notch, $T_2$, and the paleobeach are illustrated by photographs from the west side of
the study area. (a) Overview showing the unconformable relationship of the Late Holocene tidal notch and the $T_2$ fan
highlighting the location of figures in other panels. (b) Oblique aerial perspective view of the outcrop with the major features
highlighted. (c) Detail of the Vermetid extraction site shows how gravels of $T_2$ overlie a Vermetid shell pocket in the tidal
notch. (d) Detail of the contact zone between the carbonaceous bedrock, $T_2$, and the tidal notch (partly buried by colluvium).
(e) The Vermetid fossil pocket is covered by $T_2$ fan material (detail of (c)).
**4.1. The Klados catchment infill units**
At ~100 m above the modern channel, a light-coloured, unsorted, and unconsolidated deposit ($L_1$) with matrix-
supported, subangular clasts crops out (Fig. 4c). No bedding or other flow indications, such as imbrication,
sigmoidal structures, or layering, are preserved within the deposit. The $L_1$ deposit shares many similarities with
rock avalanche deposits described elsewhere by Dufresne (2017), which is why we interpret the deposit's
observable parts as the body facies of a landslide. The carapace and basal facies are not observable and may have
been locally eroded or buried by stream and hillslope geomorphic processes following landslide deposition. The
deposit is present along the gorge's walls up to the headwaters, where it locally backfills the pre-existing bedrock
topography in paleo-tributaries.

The alluvial fill units ($T_1$, $T_2$, and $T_3$) each consist of a coastal fan delta and its equivalent terrace upstream in the
gorge. These units consist of unconsolidated, matrix-rich but grain-supported, subangular to subrounded carbonate
boulder- to silt-sized clasts (Fig. 4d, e). At the outcrop-scale, the clasts exhibit a crude fining upward trend from
a coarse, unsorted, angular to subangular, matrix-rich basal association of boulders, cobbles, and pebbles to meter-
scale beds of moderately sorted cobbles, pebbles, and sand. This vertical variability is consistent for all the
significant terraces such that the mean grain size and structure do not change between the infill deposits, except
for occasional fine-grained lenses towards the top of the two highest coastal units ($T_1$ and $T_2$) (Fig. 6). The upper
portions of the alluvial fill units are always layered and fluvially reworked, resembling the planar beds typical of
flow in shallow channels (Fig. 4d, e; c.f., Blair and McPherson (2015)). Soils are weakly developed on all three
alluvial fill units based on soil redness (or lack thereof), depth, density, and vegetation cover (Fig. S5). Moreover,
there are no discernable secondary carbonates or other mineral diagnostic horizons related to migration processes,
and pedogenic clay formation is insignificant. The soils lack fluvic properties and are well-drained, which is why
the best categorisation appears to be a calcaric, skeletal Regosol (IUSS Working Group WRB, 2015).

The $T_1$ basal layer is not exposed near the coast; however, the $T_2$ basal layers exposed adjacent to the modern
channel show laterally changing grain sizes and structure. With increasing vertical distance to the channel thalweg,
$T_2$ grades from a matrix-rich, unsorted association containing various grain sizes (boulders to sand) to layers
dominated by smaller clasts and increased sorting (Fig. 6). The grain size distribution and structural observations
suggest that the units' stratigraphically-lowest deposits correspond to debris flows buried by an increasing amount
of braided river deposits.

The paleo-beach deposit at 6 m a.s.l. has similar sedimentology to the modern beach and consists of cemented,
clast-supported layers of rounded, discoidal carbonate sand, pebbles, and small cobbles (Fig. 4b). Two units
overlie the paleobeach, the $T_2$ basal debris flow crops out to the west, and an intermediate deposit characterised
by smaller grain sizes and lobate structures ($D_{LOB}$) in the east (Fig. 4f). $D_{LOB}$ most likely consists of the talus cones
formed during the erosion of the $T_1$ cliff before $T_2$ deposition.

The $T_1$ fan is locally topped by a discontinuously laminated, homogeneously reddish silty clay (unit ED; Figs. 2;
4g). It contains a few angular pebbles (~1 cm) and terrestrial snail shells. The material is interpreted as wind-
transported sediment commonly found in the coastal areas of Crete (Ott et al., 2019b). Deposition and preservation
of this eolian deposit commenced after the abandonment of the $T_1$ surface during the incision phase. Thus, the $T_1$-
ED unconformity combined with the radiocarbon measurements of the shells in the ED deposit constrains the
onset of incision into the $T_1$ deposit.

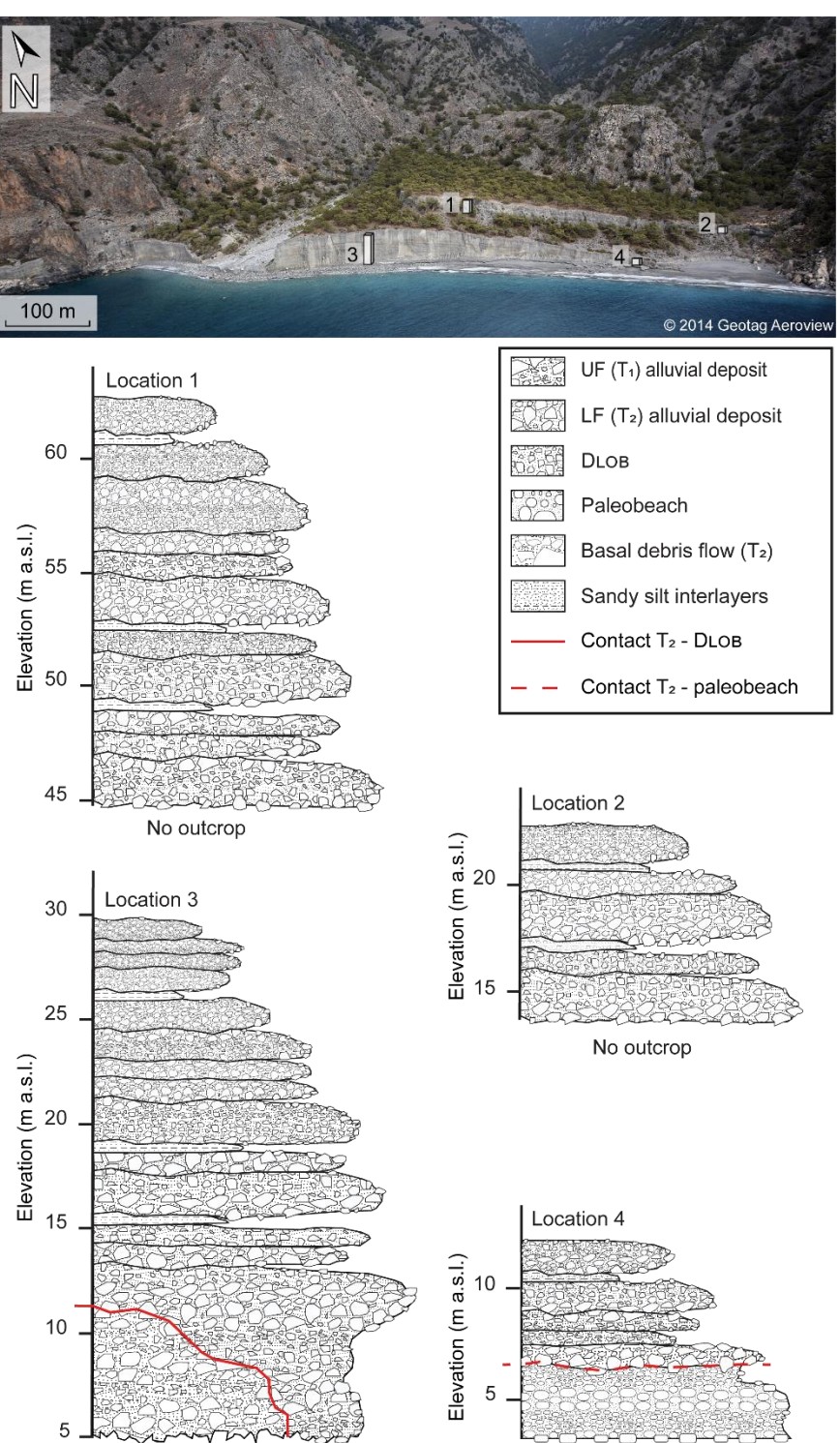

**Figure 6**: Stratigraphic sections of the alluvial fans. The section locations are highlighted in the top panel.

## 4.2. Cross-cutting stratigraphic relationships reveal the relative sequence of events

Stratigraphic contacts between units provide us with a relative sequence of events and a possible timeframe for the individual aggradation and incision phases. Inset relationships and buttress unconformities allow for a relative chronology for $L_1$ and the alluvial deposits ($T_1$, $T_2$, and $T_3$). $T_1$ is inset into and forms a buttress unconformity with $L_1$ (Fig. 4a), and the same relationship exists between $T_1$ and $T_2$), and $T_2$ and $T_3$ (Figs. 2; 5a). These cross-cutting relationships show that the valley-filling units, arranged in age from oldest to youngest, are $L_1$, $T_1$, $T_2$, and

$T_3$. Note that this sequence requires repeating episodes of valley-wide aggradation and incision. In the lower parts of the valley, incision cuts into pre-existing fill forming a buttress unconformity, while farther up the valley, incision persists through the pre-existing fill and into bedrock, generating bedrock straths. We also note that L1 fills in a pre-existing bedrock paleo-topography.

The most important cross-cutting relationship observed in the field was a buttress unconformity between the $T_2$ alluvial fan delta and Late Holocene bio-erosional notch, which we refer to as the Krios notch due to its association with the Krios paleoshoreline (Fig. 5a-d). Along the western end of the study area, the $T_2$ fan buries the Krios notch (Fig. 5a, c, d). Furthermore, we observed that angular $T_2$ gravels locally buried Vermetid and sea urchin fossils adhered to the notch and we sampled them for radiocarbon dating (see below) (Fig. 5c). This observation demands that the paleo-sea level marker was carved before $T_2$ deposition. Finally, the paleobeach deposit is horizontally aligned with the tidal notch, and while $T_2$ unconformably overlies both, the paleobeach also shows an unconformable contact to $D_{LOB}$ (Fig. 4f). This observation suggests that the tidal notch represents different facies of the same Holocene shoreline and that both are buried by $T_2$ (Fig. 4b, f). Collectively, these observations indicate the notch and paleo-beach deposit at the Krios paleoshoreline and conclusively demonstrate that the $T_2$ deposit was emplaced in the Holocene.

Another important cross-cutting relationship is the contact between $T_1$ and the eolian deposit (ED) because this represents a minimum age for the end of the $T_1$ aggradation phase. The sharp, undisturbed contact points to a rapid shift from aggradation to $T_1$ surface abandonment. The radiocarbon ages of terrestrial snail shells from the eolian deposit provide a minimum age for this fan surface abandonment.

**4.3. Upvalley trends of the alluvial infill deposits**

The mapped unconformities between alluvial units indicate that each valley infill was deposited in a separate aggradation event, followed by a phase of incision during which the river adjusted its slope, eventually reaching and incising the bedrock in the valley's narrow reaches. The structure and sedimentology of the infill terrace deposits change vertically from unsorted debris flow deposits at the bottom to layered planar beds and regular riverine deposits at the terrace tread (Fig. 6). These trends are consistent along the river channel from the mouth to the headwaters. Thus, the initiation (trigger), transport, and deposition processes were very similar during each aggradation phase, while the onset of incision between aggradation intervals acted as the turning point that allowed the system to repeat the cycle. However, the elevation at which we find the terraces increases upstream because we find straths and the bottom contacts of the infills in the headwaters, both of which are generally absent in the lower reaches. We can utilise these bottom contacts to reconstruct the then-active river channel, which steepened upstream faster relative to the modern profile (Fig. 3). The steeper paleochannel gradient illustrates that the headwater reaches were subjected to greater incision relative to the outlet due to sediment overload. The subsequent aggradation events were fed from the landslide deposits upstream, resulting in the headwater transfer of sediment and channel slope adjustment.

## 4.4. Radiocarbon dating of shells

The radiocarbon dating of Vermetid shells constrains the timing of uplift for the Krios paleobeach and notch. The ages are reported in radiocarbon years (yr) and fraction modern (fm) (Table 1). The samples, collected at and below the notch (Fig. 2), yielded radiocarbon ages of $2,672 \pm 24$ [14]C yr BP (2,844-2,300 cal yr BP) and $2,397 \pm 25$ [14]C yr BP (2,116 – 1,941 cal yr BP) for the uplift, respectively. These ages are slightly older than the inferred age of uplift in the first centuries AD. The difference might be caused by the organisms' death before the uplift due to natural causes or burial by the prograding fans. A terrestrial gastropod shell collected from the ED deposit capping $T_1$ yielded an age of $3,952 \pm 24$ [14]C yr BP (4,407-4,157 cal yr BP), thereby constraining the cessation of $T_1$ aggradation.

Table 1: Radiocarbon measurements from the Klados catchment. Results are reported in fraction modern (fm) and radiocarbon years (yr). All samples were analysed at the ETH radiocarbon lab except for #3, was analysed by Direct AMS, Bothel, WA, USA.

| Label | Lab ID | Deposit | Sample type | Coordinates | Fraction modern ± error absolute | [14]C age (1 σ) ± error (yr) |
|---|---|---|---|---|---|---|
| 1 | 82442.1.1 | Tidal notch | Vermetid carbonate shells | 35.2295 N 23.9093 E | $0.717 \pm 0.00213$ | $2,672 \pm 24$ |
| 2 | 85020.1.1 | Tidal notch | | | $0.742 \pm 0.00234$ | $2,397 \pm 25$ |
| 3 | D-AMS 011054 | Eolian deposit (ED) | Terrestrial snail | 35.2366 N 23.9159 E | $0.6114 \pm 0.0018$ | $3,952 \pm 24$ |
| 4 | 94494.1.1 87102.1.1 | Tributary (TD$_1$) | Bulk sediment | 35.2320 N 23.9155 E | $0.573 \pm 0.00690$ $0.711 \pm 0.00695$ | $4,820 \pm 556$ $2,696 \pm 369$ |
| 5 | 94495.1.1 87100.1.1 | Tributary (TD$_2$) | Bulk sediment | 35.2320 N 23.9155 E | $0.566 \pm 0.00690$ $0.663 \pm 0.00657$ | $4,820 \pm 379$ $3,389 \pm 587$ |
| 6 | 94493.1.1 | Lower fan (T$_2$) | Bulk sediment | 35.2308 N 23.9125 E | $0.583 \pm 0.00680$ | $4,793 \pm 826$ |
| 7 | 94491.1.1 87103.1.1 | Upper fan (T$_1$) | Bulk sediment | 35.2266 N 23.9156 E | $0.531 \pm 0.00650$ $0.613 \pm 0.00714$ | $5,788 \pm 874$ $4304 \pm 903$ |
| 8 | 87103.1.1 | Upper fan (T$_1$) | Bulk sediment | 35.2275 N 23.9140 E | $0.579 \pm 0.00602$ | $5,131 \pm 1,342$ |
| 9 | 94496.1.1 87098.1.1 | Landslide (L$_1$) | Bulk sediment | 35.2309 N 23.9186 E | $0.728 \pm 0.00740$ $0.671 \pm 0.00737$ | $2,476 \pm 351$ $3,294 \pm 831$ |

## 4.5. Radiocarbon dating of alluvial infill deposits

Sedimentation ages were constrained by bulk radiocarbon dating, and are reported as radiocarbon ages ([14]C yr BP). The corresponding fraction modern (fm) values are specified in Table 1 and Fig. 7. For the alluvial deposits, samples were collected from fine-grained slack-water lenses at the top of each deposit (Fig. 2; 6). The $T_1$ samples returned ages of $5,788 \pm 874$, $4,304 \pm 903$, and $5,131 \pm 1,342$ [14]C yr BP. The low carbon content causes age uncertainty in the samples, and thus, the ages need to be interpreted with care. Due to the inaccessibility of fine-grained lenses on the seaward cliff of the $T_2$ deposit, no samples were collected. However, a sample collected from just below the tread and close to the apex of the $T_2$ fan yielded an age $4,793 \pm 826$ [14]C yr BP [14]C yr BP, respectively. Two samples (TD$_{1\ \&\ 2}$) from a fine-grained deposit at the confluence of the major tributary and the Klados River yielded ages of $4,820 \pm 556$ and $2,696 \pm 369$ (TD$_1$), and $4,820 \pm 379$ and $3,389 \pm 587$ (TD$_2$), respectively. These ages confirm that at least one of the valley infills reached 40 m elevation upstream of the fan

apex and blocked the tributary channel. The landslide deposit ($L_1$) returned ages of 2,476 ± 351 and 3,294 ± 831
$^{14}$C yr BP.

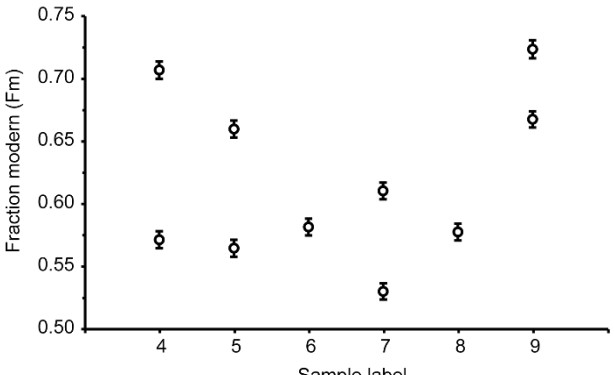


**Figure 7**: Fraction modern (fm) radiocarbon data of bulk sediment samples from the major deposits. Detailed information on
the measurement is reported in Table 1. The data are presented with an absolute error. Sample labels 4 & 5= $TD_{1\ \&\ 2}$, 6 = $T_2$,
7 & 8 = $T_1$, 9 = $L_1$.
The radiocarbon data presented here demonstrate that the deposition of the Klados sedimentary deposits occurred
during the mid-to-late Holocene. However, the order of some of the resulting ages is inconsistent with the relative
sequence of events demanded by observed cross-cutting relationships. This mismatch is likely due to the
admixture of organic carbon from different sources and the finite amount of measurable material recoverable from
the samples. We discuss these sources of uncertainty in detail in the discussion below and base our further
interpretations on the 1-σ uncalibrated radiocarbon ages.
**4.6. Volumes of rockfall and valley infill**
The thicknesses of the potential source area rockfall slabs were estimated based on friction angles and the extent
of the planar surfaces on Volakias Mtn. at the head of the Klados catchment (visualised in Figs. 8; S3). The
calculation resulted in six downward converging wedges between 2.8 x $10^5$ m$^3$ and 3.82 x $10^9$ m$^3$ (Table 2). The
estimated volume of the valley infill, calculated as described above, is between 9.37 x $10^6$ m$^3$ and 3.05 x $10^7$ m$^3$.
(Fig. 8). These volumes lie within the range of our upper-intermediate wedge volumes and values reported for
similar size events in previous studies (USGS, 2016). Thus, we expect volume estimates to be relatively robust.

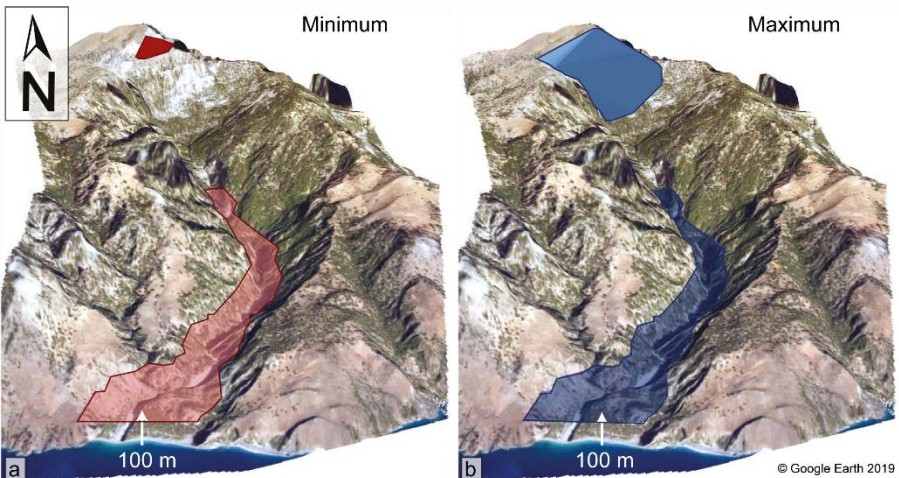


**Figure 8**: Visualisation of the minimum (a) and maximum (b) rockfall volume as calculated for the dynamic landslide runout model, and the highest possible reference plane inferred from the location of the highest mapped deposits (100 m above the modern river channel; Fig. 3a).

**Table 2**: The maximum and minimum volume estimations for the rockfall, the valley infill, and the fan material. We calculated six downward-converging wedges of different volumes for the initial rockfall volume. The intermediate wedge volume was used for the landslide runout modelling.

| | Minimum [$m^3$] | Maximum [$m^3$] | Intermediate [$m^3$] |
|---|---|---|---|
| Rock fall (wedges) | $2.80 \times 10^5$ | $3.82 \times 10^9$ | $8.34 \times 10^8$ |
| | | | $2.23 \times 10^7$ |
| | | | $9.08 \times 10^6$ |
| | | | $9.18 \times 10^5$ |
| Valley infill (landslide) | $9.37 \times 10^6$ | $3.05 \times 10^7$ | |
| Fans alone | $5.30 \times 10^5$ | $1.15 \times 10^6$ | |

## 5. Discussion

### 5.1. Timing of the Klados catchment stratigraphy from relative and absolute dating

The agreement between our field observations and radiocarbon geochronology strongly supports a Holocene age for the alluvial infills in the Klados catchment. Despite their large uncertainties, all the radiocarbon measurements are Holocene and are mostly consistent with the observed cross-cutting field relationships. Even without the radiocarbon data, the following series of observations indicate geologically recent emplacement of the Klados alluvial fill units. Firstly, the buttress unconformity between the $T_2$ and the late Holocene Krios paleoshoreline requires a post-late Holocene deposition of $T_2$. Secondly, the immature soils developed on $T_1$ and $T_2$ surfaces are inconsistent with the well-developed Bk and Bt horizons on Pleistocene alluvial fans with similar parent rock source areas (Fig. S5; Gallen et al., 2014; Pope et al., 2008). Thirdly, the slopes of the $T_1$ and $T_2$ surfaces match the modern channel slope in the lower reaches, suggesting that the paleoriver prograded to a base level similar to the modern sea level. Finally, the morphology of the coastal cliff on $T_1$ and $T_2$ is similarly (Fig. 1e; also Fig. 4a of Mouslopoulou et al., 2017)). Given that both infills are unconsolidated sediment, one would expect relatively rapid slope degradation (diffusion) of the paleo-sea cliff as noted along scarps produced by fault rupture (Nash, 1980)in similar sedimentary deposits. Indeed, scarp diffusion modelling suggests that a Holocene age of the $T_1$ paleo-sea cliff provides a more reasonable approximation of the diffusion coefficient, considering climate and material properties than a Pleistocene age based on a comparison with a recent global compilation of diffusion coefficients (Fig. S6; Richardson et al., 2019).

The relative age control on the landslide deposit ($L_1$) does not require a Holocene emplacement. However, due to this deposit's high erodibility, it is unlikely to persist in this landscape for an extended period, and soil development on this deposit is relatively immature. These findings, coupled with the Holocene bulk radiocarbon ages from the $L_1$ deposit, lead us to conclude that the landslide deposit is also Holocene.

Our radiocarbon ages are exclusively Holocene, but bulk radiocarbon measurements will introduce uncertainties to the chronology. Sources of error are diverse and closely related to environmental variables. At Klados, a decrease in measured age relative to real age most likely originates from the secondary incorporation of recent organic matter, while the inclusion of radiocarbon-dead bedrock carbonates causes an overestimation. Both of

these sources of error are minimised in our approach. On the one hand, recent organic matter is often associated
with large grain size fractions (c.f., Rothacker et al., 2013) and is easily avoided during sample collection and
preparation. Conversely, the potential of sample age overestimation is minimised by fumigating the samples
before measurement. This step ensures the substantial removal of inorganic carbonate. An uncertainty unrelated
to the environment is introduced by the low TOC and can result in smaller sample sizes prepared for the bulk
radiocarbon measurement, which were affected with larger uncertainties after corrections for processing blanks
and standards (Ruff et al., 2010). Nevertheless, empirical studies show that samples that contain a mixture of
young and old carbon may overestimate the age of a deposit by 500–2000 years (Grimm et al., 2009; Rothacker
et al., 2013). We recognise that the bulk sediment dating results contain inherent uncertainties and express
reference timeframes rather than absolute ages for the processes due to a possible overestimation of the age.

Our Holocene radiocarbon ages complement field observations and provide additional age control. Except for one
outlier, the deposition order obtained from the radiocarbon dating agrees with the sequence of events established
in the field. The valley infill $T_1$ predates $T_2$, which is approximately the same age as the main tributary's slack-
water deposits. The only outlier to the sequence is $L_1$. However, the clear stratigraphic relationship between $L_1$
and the other deposits overrules the radiocarbon dating. The most likely cause for the radiocarbon age discrepancy
is the introduction of younger organic matter after $L_1$ deposition by erosional processes, water movement, or
bacterial activity. Consequently, both radiocarbon age dating and field observations imply the geologically recent
deposition of the Klados stratigraphic sequence. Both the bulk sediment radiocarbon ages and the radiocarbon
ages from shells are consistent with a Holocene age for all deposits. The cross-cutting relationships allow for a
precise relative chronology of events during a relatively short amount of time.

The Holocene age for the Klados alluvial deposit sequence proposed here differs significantly from previous
dating results by infrared stimulated luminescence (IRSL) on feldspar (Mouslopoulou et al., 2017), which resulted
in Pleistocene ages for the infills (29-50 kyrs BP). However, the field observations and cross-cutting relationships
demonstrate that these deposits are Holocene, supported by our radiocarbon analyses. Luminescence burial dating
of deposits exploits the assumption that charge is gradually built up in feldspar or quartz grains due to radiation
from radiogenic decay of radioactive elements and cosmic rays. To relate the amount of charge a grain releases
as luminescence signal to the duration of sediment burial (depositional time of unit), all charge within the crystal
lattice needs to be fully released by sun bleaching before deposition; a process that requires seconds of full sun
exposure for quartz and minutes for feldspar (Rhodes, 2011). Alluvial fans, especially in small catchments with
short transport and a significant portion of debris flow deposits, are therefore prone to biases in luminescence
measurements because the short transport in sediment-rich flows usually does not allow for a complete bleaching
of the mineral grains, and especially not feldspar (Rhodes, 2011). This effect is enhanced because minerals freshly
released from the bedrock have worse luminescence characteristics and take longer to bleach (Rhodes, 2011).

The anomalously old luminescence ages reported by Mouslopoulou et al. (2017) are likely biased due to
incomplete bleaching caused by the turbulent mode of transport (Rhodes, 2011). The broad positively skewed age
distributions of measured equivalent dose measurements (the amount of charge released from the grains) in
Mouslopoulou et al. (2017) from feldspar IRSL indicate a mix of bleached and unbleached grains resulting in late
Pleistocene ages for both fan units. The mixture of bleached and unbleached grains is especially evident because
Mouslopoulou et al. (2017) also measured the quartz OSL signal, and found the same positively skewed age
distributions but with younger ages. The discrepancy between the younger quartz OSL and older feldspar IRSL
measurements can be explained by the more rapid bleaching of quartz grains; however, these authors discarded
and did not report the OSL ages choosing instead to construct their interpretation on the IRSL measurements
alone. Furthermore, the Pleistocene luminescence ages are difficult to reconcile in the context of similarly
immature soil development and similarly crisp cliff morphology among deposits that are reported as greater than
30 ka with ~10 kyrs separating the emplacement of each unit. For these reasons, we consider the IRSL dates from
Klados as biased and not representative of the accurate depositional ages of the alluvial fans. Instead, our data and
field observations are only consistent with a Holocene age of the Klados alluvial fill deposits.
An important implication of the finding of Holocene ages for the stratigraphic units in the Klados catchment is
that within short periods, the catchment alternates between phases of valley-wide aggradation followed by
intervals of rapid incision through the valley fill and into the bedrock in the upper portions of the catchment. This
stratigraphic history is distinct relative to adjacent catchments that record slower and steadier aggradation and
incision histories (Pope et al., 2008, 2016). This evidence indicates that local and unique processes in Klados are
responsible for high-frequency pulses of aggradation and incision. We hypothesise that the large landslide deposit
($L_1$), the oldest unit identified in the catchment, is the sediment source for the younger, inset alluvial fans. This
inference is supported by our volume reconstructions and is in large part responsible for the unique stratigraphy
and geomorphic evolution of Klados. We explore this hypothesis in detail below.
**5.2. A rockfall source for Holocene deposits in the Klados catchment**
Most of the adjacent gorges have alluvial infills, but these do not reach the thickness of the Klados deposits, and
to date, only one other case study shows thick Holocene deposits. These are reported in the Aradena Gorge 10 km
west of Hora Sfakia, where alluvial terraces up to 14 m above the modern channel bed are preserved (Maas and
Macklin, 2002). They aggraded upstream from channel reaches temporarily blocked by landslide deposits and
were incised in the next high-intensity discharge event. The authors dated the deposits to the last 200 years using
lichenometry and dendrochronology (Maas and Macklin, 2002). Even though the Aradena Gorge's deposits only
span the last 200 yrs., the over-proportionate amount of sediment in the system and the rapid aggradation and
incision rates recall the situation in the Klados Gorge where the large amounts of sediment are without an apparent
source, and up to 30 m thick deposits of $T_2$ aggraded after the first centuries AD. Moreover, the alternating narrow
and wide sections in the Klados Gorge (Fig. 2) are prone to blockage, which may control sediment distribution
and terrace genesis in a fashion similar to that reported by Maas and Macklin (Maas and Macklin, 2002). Both the
Klados and the Aradena River deposits require rapid sedimentation rates beyond what is commonly reported, and
the only possible explanations require local, isolated sources of sediment.
To elevate the sedimentation rates in the Klados Gorge, but not in adjacent gorges, and to enable the aggradation
of such large volumes so quickly, an extraordinary but spatially limited sediment input is required. We hypothesise
that a massive rockfall in the headwaters of the Klados River provided the necessary amount of loose sediment
and the impetus for aggradation and incision. The rockfall hypothesis is the most straightforward option because
the Klados catchment's hillslopes are supply-limited, mantled by only a thin layer of regolith. Furthermore,
cosmogenically-derived erosion rates from nearby catchments with comparable rock-types suggest erosion rates
on the order of ~0.1 mm yr$^{-1}$ (Ott et al., 2019a), which is too low to generate the observed volume of detritus over
Holocene timescales. Moreover, its unconsolidated state makes a landslide the ideal source material.
**5.3. Landslide runout modelling supports the hypothesis of landslide sediment origin**
Field observations document the presence of landslide deposits scattered throughout the valley. The spatial extent
and sedimentology of this the L1 deposit are consistent with the pulverised remnants of a high-magnitude rockfall.
The likely source for this rockfall is the cliff face at the headwaters of the Klados catchment, which has
overhanging rock bands that are diagnostic of recent rockfall events (Figs. 1b; 3e). Below these overhanging
features, a steep (> 35˚) planar bedrock slope abruptly terminates at the Klados catchment floor (Fig. 1a). We
hypothesise that a large rock mass detached from the upper portion of this mountainside, dropped to the catchment
floor, pulverising upon impact and backfilling the paleo-valley. We envision a process similar to large rock falls
with extended drop heights that have been observed in places like Yosemite Valley, CA (Wieczorek et al., 2000)
and the Swiss Alps (Mergili et al., 2020).
To evaluate a large valley-filling landslide hypothesis, we use the landslide runout model DAN3D-Flex (see
details in the supplement). DAN3D-Flex allows for an initial stage in which the rockfall slides as a single, coherent
mass to simulate a rockfall with minimal disruption (Aaron and Hungr, 2016). The flow behaviour of the landslide
is defined based on parameters from back-analyses, rock properties, and the input topography (in this case the 5
m DEM). We ran a suite of different parameter combinations to find the best-fit runout, as defined by the final
landslide extent and thickness compared to the mapped landslide deposit. Only the best-fit model is presented
here for brevity, but details on the model runs can be found in the supplement.
The best-fit simulated runout for the landslide was obtained after multiple runs using inputs as defined in Table
3. Static images extracted from the model runout show the landslide's position and extent at selected time intervals
(Fig. 9). We constrained the initial gravitational movement to 30 seconds, after which the rockfall reaches the
bottom of the sliding plane, hits the valley floor, and disintegrates (Fig. 9b). The rheology controlling flow
behaviour changes upon impact from rigid body frictional to Voellmy-rheology controlled granular flow,
consistent with our hypothesis that the rock mass pulverised upon impact with the valley floor. At about 45
seconds after initiation, the landslide reaches the sharp bend in the valley, slows down, and bulks up by the vertical
concentration of the mass in the head of the landslide (Fig. 9c). After 60 seconds, the landslide is obstructed by
the cliff in the centre of the modern fan structure (Fig. 9d). The modelled failure mass bulges up and overflows
the cliff (Fig. 9e, f). No high-resolution bathymetry data were available, which precluded offshore runout
modelling. The average deposit thickness is highest in the centre of the channel and decreases with distance to the
impact site because entrainment was forbidden while the loss of material by deposition was implemented in the
model. The averaged thicknesses at the final model time-step correspond well with the elevations of the landslide
deposit in the field of ~100 m. The narrow sections of the valley did not obstruct the flow too much from the
model outputs, which suggests an even distribution of the landslide material.

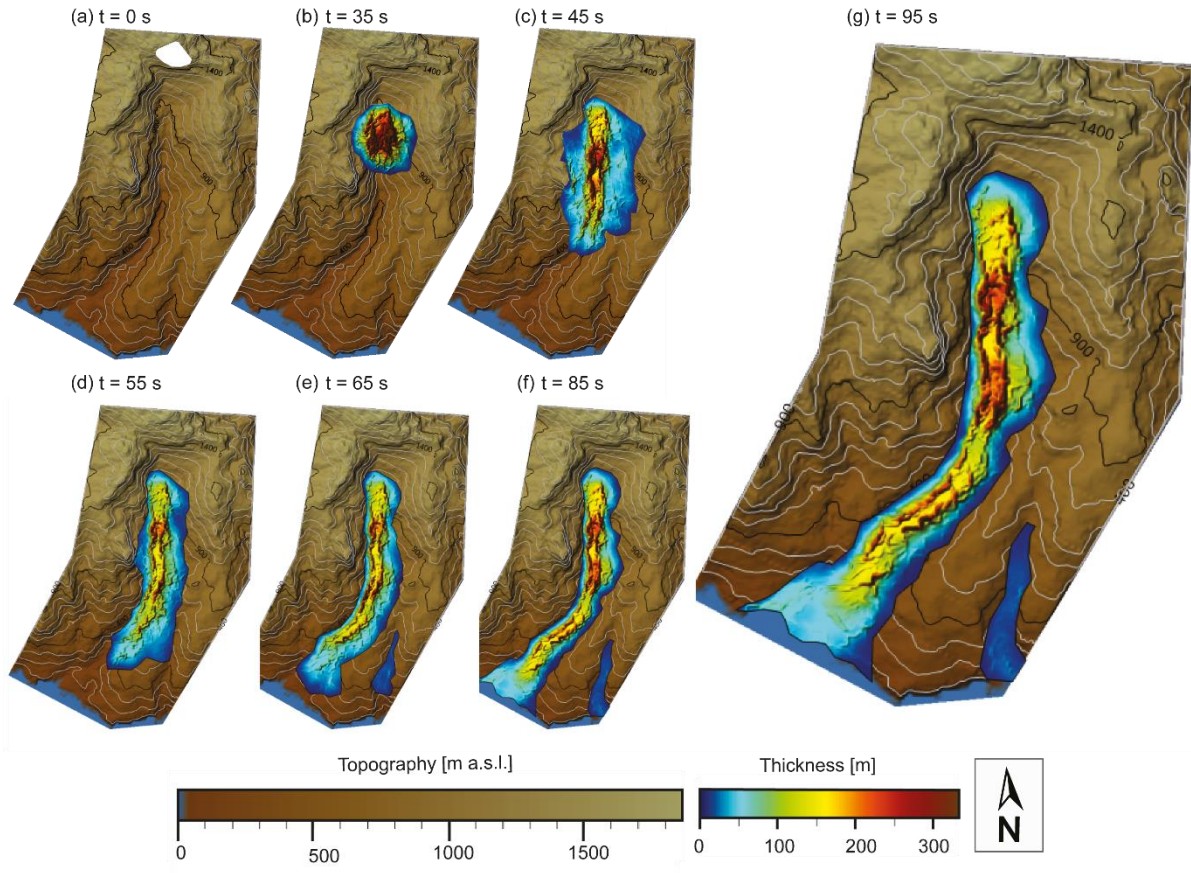

(a) t = 0 s   (b) t = 35 s   (c) t = 45 s   (g) t = 95 s

(d) t = 55 s   (e) t = 65 s   (f) t = 85 s

Topography [m a.s.l.]

0   500   1000   1500

Thickness [m]

0   100   200   300

N

**Figure 9**: Time slices from the landslide runout modelling with an intermediate rockfall volume ($9.08 \times 10^7$ m$^3$). The rockfall moved as an intact block from the mountainside but partly pulverised and liquefied upon impact with the valley floor. It partially infilled the valley as a ground-based landslide and a cloud of dust. The landslide eventually reached the sea, leaving scattered deposits up to 200 m thick in the valley centre. No major blockage is shown in the narrow valley reaches. The model indicates that part of the landslide crossed into the adjacent valley, but this part of the coastline was not investigated during fieldwork.

**Table 3**: Parameters for the five best-fit landslide runout models. The quality of correlation between the model and field observations decreases with increasing numbering (1-5).

| Quality of correlation | 1 | 2 | 3 | 4 | 5 |
|---|---|---|---|---|---|
| Rheology | Voellmy | Voellmy | Frictional | Frictional | Frictional |
| Input volume [km$^3$] | 0.0908 | 0.8335 | 0.00908 | 0.8335 | 0.00908 |
| No of particles | 2000 | 2000 | 2000 | 2000 | 2000 |
| Time steps [s] | 0.1 | 0.1 | 0.1 | 0.1 | 0.1 |
| Velocity smoothing coefficient | 0.02 | 0.02 | 0.02 | 0.02 | 0.02 |
| Stiffness coefficient | 200 | 200 | 200 | 200 | 200 |
| Rigid behaviour time [s] | 30 | 30 | 10 | 10 | 30 |
| Unit weight [kN m$^{-3}$] | 21.5 | 21.5 | 20 | 20 | 20 |
| Internal friction angle | 35 | 35 | 15 | 15 | 20 |
| Friction coefficient | 0.2 | 0.2 | 0 | 0 | 0 |
| Viscosity [kPa s] | 0 | 0 | – | – | -, 1000, - |
| Turbulence coefficient [m s$^{-2}$] | 500 | 500 | 0 | 0 | 0, 500,0 |
| Internal friction angle | 35 | 35 | 35 | 35 | 35 |

| | | | | | |
|---|---|---|---|---|---|
| Maximum slide velocity [m s$^{-1}$] | 213.7 | 308.9 | 658.5 | 770.4 | 246.3 |
| Travel time [s] | 99.8 | 107.6 | 85.4 | 94.2 | 34.4 |


The best-fit runout model agrees particularly well with our field observations of landslide deposit thickness and
the resulting valley infill's extent. Additionally, we discarded models with maximum slide velocities greater than
the speed of sound, and travel times of less than 1 minute (Table 3). The model results show that the material
moved through the valley and was deposited at a sufficient thickness and elevation above the paleochannel to
explain the deposits we identified in the field. Furthermore, the modelled flow rheology is consistent with the
observed deposit sedimentology. The dominance of fine grains can be explained by the initial rockfall evolving
into two modes of transport upon impact with the valley floor. If correct, the impact with the valley floor
transformed the rockfall into a partly "liquefied" landslide due to air inclusion and abrasive grain interaction, but
also a wind-blast driven sand-cloud which reached several hundred meters elevation (c.f., Wieczorek et al., 2000).
Fluvial reworking mainly affected the landslide's coarse-grained parts in the valley while the finer-grained portion
remained on the catchment walls. Nevertheless, the two modes of transport might explain the high amount of fine
material in the subsequent alluvial deposits, as they were sourced from both of the landslide deposit types. Our
model offers a first insight into the initial rock fall's behaviour, the location of the material brought from the
mountain face, and supports the initial hypothesis of a landslide as the source material for the younger deposits in
the valley. We discuss further in-depth caveats on the modelling in the supplement sect. 2.
**5.4. Stochastic versus external forcing for aggradation-incision cycles**
The alluvial deposits in the Klados catchment are volumetrically oversized and immature in soil development
compared to other catchments in southern Crete. We have demonstrated that the deposits preserved in the valley
are Holocene in age and that following a massive landslide event, the catchment dynamics are best described by
rapid and dramatic alternations between valley-wide aggradation and incision. These findings show that the
emplacement of the landslide deposit altered catchment dynamics, making Klados more sensitive to external
forcing. This change in sensitivity to external forcing makes the Klados fans distinct among the well-studied
Pleistocene fans in Crete. While in each case, sediment transport events are likely associated with high-intensity
rainstorms, as indicated by the high-energy depositional environments inferred from fan stratigraphy in Klados
and Pleistocene fans elsewhere on Crete, the threshold magnitude for a sediment-generating event, whether  a
rainstorm or seismically-driven ground shaking, in Klados is likely much smaller relative to those that produced
the Pleistocene fans. This difference in sensitivity to external forcing makes the Klados fans unique in the context
of Pleistocene fans of Crete. Furthermore, as detailed below, the available evidence shows that alluvial fan and
terrace development in Klados is a transport-limited process, whereas Pleistocene fan construction on Crete is
commonly supply limited.

Alluvial terrace and fan formation are fundamentally driven by variations in the ratio of sediment and water
discharge rates. Studies of Pleistocene coastal alluvial fans sequences on Crete show that fan deposition is roughly
coincident with cooler glacial or stadial periods or the timing of transitions in climate (e.g., cool to warm or vice
versa) (Gallen et al., 2014; Pope et al., 2008, 2016; Runnels et al., 2014; Wegmann, 2008). Studies demonstrate
that precipitation rates, and thus water supply, across the eastern Mediterranean basin apparently do not fluctuate

dramatically during late Quaternary glacial-interglacial cycles (Hijmans et al., 2005; Watkins et al., 2018); although effective moisture and flashiness of precipitation-discharge events are likely different between stadials and interstadials. This evidence implies that Pleistocene alluvial development on Crete is primarily a function of climate-modulated variations in sediment supply rate; alluvial fans form when hillslope sediment production rates are higher than the present, which is not surprising given that most hillslopes in Crete expose large amounts of bare bedrock draped with thin patchy sediment suggesting supply-limited conditions. A reasonable interpretation is that more active periglacial processes such as frost or subcritical cracking generate larger volumes of hillslope sediment relative to contemporary conditions during cooler intervals. This interpretation is supported by observations of active normal fault scarps throughout Crete; those with Holocene rupture expose steep, crisp, polished fault scarps while higher, older positions of the fault plane exposed before the Holocene are more gently sloping and degraded (Caputo et al., 2006; Mouslopoulou et al., 2014). Consistent with interpretations of causes for similar active fault scarps observed in the Central Apennines, Italy, this morphology is interpreted as the result of climate-related changes in physical weathering and hillslope sediment production rates (i.e., Tucker et al., 2011). This interpretation suggests that typical Pleistocene fans in Crete are primarily a function of sediment availability (e.g., supply limited) and consist of sediment largely derived from physical weathering. This process helps explain the generally coarser grained detritus that comprises most Pleistocene alluvial fans in Crete relative to those observed in Klados. We note that this inferred general climate-driven mechanism of Pleistocene alluvial fan development on Crete is consistent with data and interpretations of many Pleistocene and Holocene alluvial fans globally (Blair and McPherson, 1994; Bull, 1991; Orr et al., 2020; Schumm, 1973; Waters et al., 2010).

In contrast to the Pleistocene alluvial fans on Crete, the Holocene Klados alluvial fan deltas and fluvial terraces result from a transport-limited process. This difference resulted from the unique conditions imposed by the deposition of the valley-filling landslide sediment that fundamentally altered the catchment-scale geomorphology and source-to-sink sediment dynamics of Klados relative to other coastal drainages in southern Crete. Before the landslide event, water discharge outpaced sediment availability and cleared the valley of loose sediment, resulting in bedrock incision; the catchment was in a sediment supply-limited state. Deposition of the highly-erodible, unconsolidated, valley-filling landslide detritus pushed the catchment into a transport-limited state. The landslide event and valley-filling sediment reduced the critical threshold for sediment mobility to the point where moderate and average rain storms or moderate-to-large regional earthquakes turned into sediment generation events that initiated sediment cascades. Due to the location of the highly-erodible landslide deposit within the valley, detritus liberated by these sediment generation events is well-connected to the river system and easily transported by subsequent rainstorms following a valley-wide triggering event. Later fluvial transport results in aggradation throughout the valley and explains the deposition of the alluvial terraces and fan deltas. Once starved of sediment, the river transport capacity increases and it is capable of incising through the alluvial fill and eventually into the bedrock near the headwaters. The change from a system whose aggradation-incision cycles thus mainly depended on sediment availability (i.e., supply limited) to a regime where sediment mobilization and transport capacity became the controlling variable (i.e., transported limited) resulted in a locally-isolated, rapid build-up of alluvial deposits.

## 5.5. Holocene evolution of the Klados catchment

Combining the relative and absolute chronology allows us to reconstruct the following history for the Holocene topographic evolution of the Klados catchment (Fig. 10). The backfilled paleo-topography preserved beneath the $L_1$ deposits indicates that the catchment was originally a bedrock-dominated channel similar to the adjacent catchments. Based on this unconformity and the sedimentology of $L_1$, the bedrock valley was instantaneously filled with unconsolidated $L_1$ sediment. The sedimentology and distribution of $L_1$ deposits are most consistent with a landslide following a large rockfall event in the catchment's headwaters. Our interpretation, reinforced by landslide runout modelling, is that the rockfall detached in the headwaters of the catchment, and upon impact with the valley floor partly liquefied and pulverised, sending debris downstream, eventually backfilling the valley up to 100 m locally, as is evidenced by preserved deposits in the high altitudes of the hillslopes surrounding the channel ($L_1$).

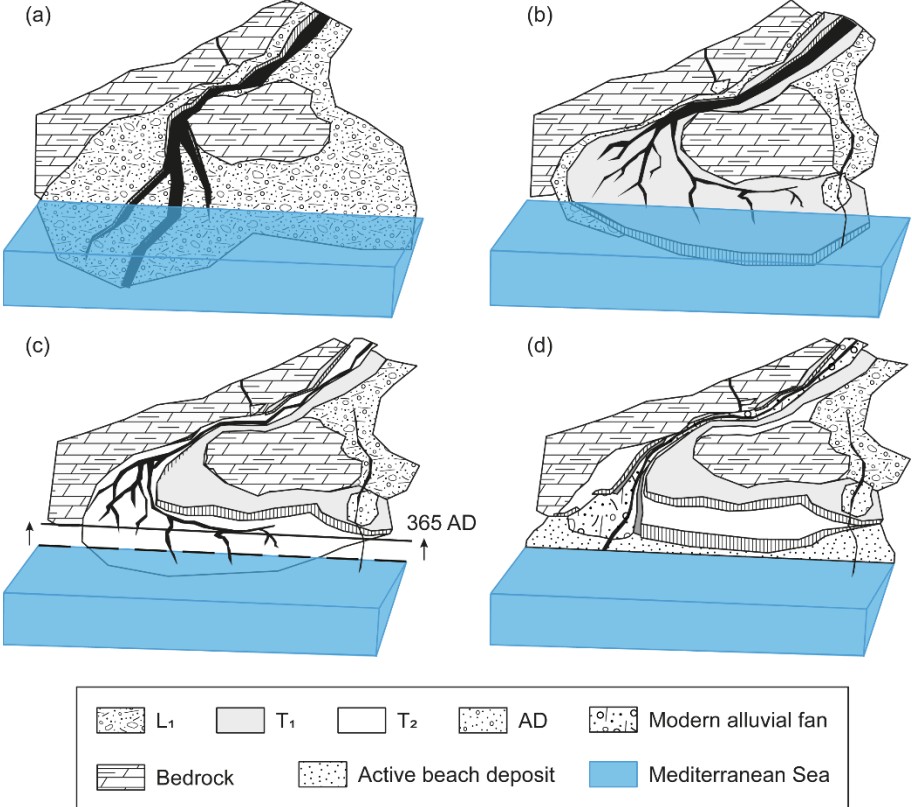

**Figure 10**: Summary of the evolution of the Holocene deposits at the outlet of the Klados catchment. Before the landslide, the valley was likely a sediment-limited, bedrock-dominated catchment similar to its modern neighbours. (a) The landslide filled the valley, which provided tools for the subsequent incision. (b) After having incised nearly to the bedrock, an unknown event caused sediment remobilisation and aggradation of $T_1$. (c) The earthquake in AD 365 uplifted the coastline by 6 m and triggered new debris flows that eventually formed $T_2$ and buried the uplifted paleobeach. (d) The modern configuration at the beach with two large inset fans burying the paleobeach.

The landslide debris deposit changed the channel's slope and altered the ratio of sediment supply and transport capacity of the Klados River by introducing vast quantities of highly erodible material throughout the valley (Fig. 10a). The river channel established a steep slope capable of transporting the imposed sediment load. Once starved of hillslope sediment supply, could incise into and through the fill deposit and into bedrock rock farther upstream near the headwaters (Fig. 3d). This relaxation of the river profile suggests a different equilibrium gradient that we

interpret was partially facilitated by tool availability in the sediment-laden channel and the stream's carrying capability. This channel relaxation period was interrupted by several episodes of valley aggradation and incision that resulted in the construction of the alluvial fill deposits, $T_1$, $T_2$, and $T_3$ (Fig 10b, c, d). Consistent with the above interpretation, field observations suggest the channel gradient steepened during the T1 and T2 aggradational phases and relaxed during the intervening incisional periods as indicated by the exposure and increasing elevations of basal strath surfaces progressively moving upstream.

Events capable of triggering debris flows and liberating vast quantities of material from $L_1$ are most likely rare, large earthquakes or large precipitation events. There is not enough evidence to discriminate between these triggering events for the $T_1$ and $T_3$ deposits. $T_2$ forms a buttress unconformity with the late Holocene sea-level notch and the uplifted paleobeach (i.e., the Krios paleoshoreline; Figs. 4b; 5). This evidence suggests that aggradation of $T_2$ may have initiated when late Holocene earthquakes, e.g., the AD 365 event, liberated sediment (Fig. 10c).

If we assume that the other large-scale aggradation events resulting in the initial depositing of L1 and subsequent $T_1$ were also caused by ground movement related to earthquake activity, we can provide a very crude estimate of the frequency of large earthquakes in the region. Assuming an early to mid-Holocene age of ~10-5 kyrs for the initial rockfall, three large events (initial landslide and two large aggradation events) took place within the last 5-10 kyrs, suggesting a crude recurrence of about 1.5-3.5 kyrs. This estimate is in good agreement with regional and more local recurrence interval estimates of great earthquakes of 800 to 4500 years (Mouslopoulou et al., 2015; Shaw et al., 2008). Of course, this is a rough estimate resting on many untested assumptions. Nevertheless, it illustrates how catchments that are sensitive to external perturbations can serve as exceptional archives, provided the geochronology and causation between catchment and external events are well constrained and documented.

Our field observations and geochronology support an alluvial fan delta and terrace formation model with four alternating phases of aggradation and incision over a geologically and geomorphically short duration (i.e., several 1000s of years). These phases are caused by the unsteady liberation of highly erodible landslide deposit material that overwhelmed the catchment and resulted in thick fans and terraces (c.f., Maas and Macklin, 2002; Scherler et al., 2016). Potential mechanisms driving the phases of aggradation and incision need to be quasi-instantaneous, and earthquakes and extreme precipitation events are the most likely options. They are capable of liberating large amounts of unconsolidated sediment in a short amount of time. However, the efficiency of erosion and transport depends not solely on the intensity of precipitation but also on autogenic processes that arise due to highly variable valley floor width in the Klados catchment. The confinement in the narrow gorge sections might have resulted in sediment damming and redistribution of material independent of external forcing. The formation and breakage of dams in the narrow bedrock reaches may have been facilitated by the random occurrence of large blocks and fluctuations in surface discharge. These would have influenced river transport capacity and sediment transport or deposition, leading to incision and aggradation by obstruction. Consequently, we argue that the substantial sediment input changed the aggradation and incision cycles in the Klados River system from those dependent on climate and sediment availability to those dependent on quasi-stochastic events such as seismicity, hillslope failures, hydraulic and sediment damming upstream from narrow bedrock reaches.

## 6. Summary and conclusions

The Klados catchment of Crete, Greece, is located in a highly dynamic bedrock landscape and features prominent inset fill terraces and associated coastal fan deltas. Our results show that the thick (> 50 m) stratigraphic sequence in the Klados catchment is Holocene in age, and we propose that a rockfall from Volakias Mountain pulverized upon impact with the valley floor and back filled a pre-existing bedrock topography with landslide debris. This interpretation is supported by the cliff face morphology, sedimentological characteristics and mapped extent of the deposit, and the results from dynamic runout modelling. The deposition of the landslide material provided a supply of highly-erodible detritus that altered catchment dynamics, leading to alternating phases of rapid valley aggradation and incision, apparently induced by seismic or high-intensity rainfall events that resulted in the construction of the impressive alluvial fill deposits throughout the Klados valley. Stratigraphy and sedimentology of the alluvial fill deposits indicate terrace and fan construction in a braided river environment. Importantly, the exposed basal stratigraphy of the younger deposits suggests a debris flow origin. Cross-cutting relationships and radiocarbon ages demonstrate that the alluvial terrace and fill sequences were emplaced in the mid-to-late Holocene; a critical and conclusive observation is that the fan associated with the second aggradation cycle forms a buttress unconformity with a Late Holocene Krios paleoshoreline 6 m above sea level typically interpreted to have been uplifted co-seismically in 365 AD. The timing of the landslide is poorly constrained but cross-cutting relationships and radiocarbon data are most consistent with deposition during the Holocene. Possible drivers for aggradation and incision cycles are fluctuations in precipitation, which influenced river transport capacity, the stochastic damming in the narrow bedrock reaches of the valley, and the increased remobilisation of sediment caused by seismic ground accelerations. Our general interpretation is that after deposition of the landslide, the catchment became ultrasensitive to external perturbations as the highly erodible landslide material lowered the threshold for a sediment-generating event. Once a significant sediment mobilisation occurred, a sediment cascade resulted in the build-up of each alluvial sequence. After the river was starved of excess hillslope sediment supply, incision commenced before the catchment was perturbed by another sediment generating event and the cycle repeated. The Klados catchment is an exceptional case study of how stochastic events can generate river terraces and alluvial fans and how particular river catchments can become hyper-sensitive to external perturbations and thereby offer the potential archiving of these external forces.

## 7. Author contribution

S.G. and K.W. planned this investigation. S.G., R.O., and E.B. are responsible for mapping and field observations.
N.H. assisted with radiocarbon analyses, K.W. provided radiocarbon measurements on terrestrial snails, E.B.
modelled the landslide runout, and V.P. advised on structure and sedimentology. E.B. prepared the samples,
analysed the results, and wrote the manuscript with contributions by all co-authors.

## 8. Competing interests

The authors declare that they have no conflict of interest.

## 9. Acknowledgements

We thank Hellenic Cadastre SA for DEM data. Sincere thanks to Christina Tsimi (NOA) for customising the 5-m
digital elevation models for the research area. We thank Jordan Aaron, who provided us with the DAN3D-Flex
software and valuable input on its usage. We thank Jonathan Booth, whose focus on the Klados catchment in his
Ph.D. thesis allowed for our accurate digital mapping.

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
