# Peer review of "Stochastic alluvial fan and terrace formation triggered by a"

_Earth Surface Dynamics, 2021_

## Community Comment (CC1)

**COMMENTS ON Bruni et al. 2021**

J. Begg, V. Mouslopoulou, D. Moraetis

6 April 2021

We are the principal authors of Mouslopoulou et al. (2017), the conclusions of which are challenged by this submission.

The Domata/Klados River area is a beautiful and under-appreciated area of Crete and this manuscript by Bruni et al. discusses the relationship between a large landslide event and deposition within a confined catchment on the southern side of the island. We believe that while the significance of the landslide event in the headwaters of the Klados River is credible, as are some of the deductions that they have made regarding its impact on deposition through the catchment, there are important elements within this manuscript that are not as straightforward as the authors have presented. We will explore some of these issues in the comments below.

**1.** The authors claim that the alluvial deposits beneath the surface T2 post-date a regional-scale earthquake in AD365 that uplifted the coastline at Klados by c. 6 m. If this is true, most of the conclusions of this work are correct. If not, however, many of their conclusions are demonstrably wrong. Thus, the authors, in our view, should have taken special care to demonstrate solidly this relationship. Below we show that they have not.

The relationship between the alluvial deposits underlying surface T2 and the AD365 "tidal notch" is not clearly presented. The authors in lines 233-234, 289-290 (and elsewhere) repeatedly claim that the AD365 "tidal notch" is overlain by alluvial deposits underlying terrace T2. However, neither Figure 4h nor 4i show this. Instead, these figures show the 365 AD tidal notch preserved on limestone bedrock (Fig. 4h) but missing from nearby gravels (Fig. 4h and 4i).

The "tidal notch" is not a deposit, it is a geomorphological feature, the result of local modification of the bedrock, here limestone, by marginal marine processes. The limestone is well lithified while the alluvial gravels are "unconsolidated" (see Section 4.3) and both lie at the inland extent of today's active beach. There is no discussion of the potential for these active marginal marine processes to erode these two lithologies differently. Would the AD365 "tidal notch", even if it had been present on the alluvial gravels (should they really be older), have been preserved? Why do they authors fail to consider this alternative scenario? The images presented do not identify the contact between limestone bedrock and "T2 deposits" (and therefore the relationship). Further, the cliff on the right-hand side of Fig. 4i comprises T2 alluvial materials and doesn't show the "tidal notch", but that does not mean that it wasn't once there before erosion by active marginal marine processes. This point is critical to the arguments, that "T2 infill deposits" (all 20 m of them) post-date the AD365 uplift event that is asserted in the rest of the paper.

Further, if the authors' interpretation above is correct: 1) the deposition of the "T2 infill deposits", 2) erosion of the lower coastal cliff and 3) incision by the Klados River below the T2 surface is required to have occurred after the AD365 earthquake. In such a scenario, the speed of deposition of the "T2 infill deposits" and their incision (by sea and river) to their present day configurations must have been exceptionally fast, with only 1600 years available to complete. Given the small catchment area and limited water flow, these events are less likely.

**2.** The unexplored problems associated with the "tidal notch" and deposition of the "T2 infill deposits" discussed above, are compounded by using their interpreted relationship to assume that the "paleobeach" deposit underlying "T2 infill deposits" must represent the AD365 shoreline. This is unproven. Instead, this correlation is based on the relationship that we questioned in (1) and on the elevation of each of the features. We argue that in our model (see Mouslopoulou et al., 2017) we would expect a "paleobeach" deposit seaward of the base of the marine cliff that truncates T1 – thus, this observation does not contradict an older age of the alluvial fans.

**3.** Reference is made by the authors to the "crisp" similarity in morphology of the two marine cliffs at Klados mouth. More careful examination of this statement shows that this is not true. The 5 m topo DEM that the authors used to derive their data is entirely adequate to contradict this assertion. See below profiles 1 to 3 across the Klados beach that illustrate that the lower sea-cliff is significantly steeper than the upper sea-cliff (75° vs. 53° average slopes). In addition, the base/crest of the lower-cliff is much sharper than those of the upper-cliff. The morphological differences between the two sea-cliffs are indicative of an age difference substantially more than 1600 years. These observations undermine the authors' assertion that the morphologies are equally immature and therefore both of late Holocene age and provide critical corroborative evidence that the upper sea-cliff is substantially older than the lower sea-cliff.

[Figure]

Figure 1 Elevation profiles at Klados River mouth, illustrating the contrasting morphologies of the upper and lower seacliffs. Profile lines are in black, but seacliffs are colour-coded, dark blue for the upper and red for the lower (note that the vertical scale = the horizontal scale). Profile locations are shown on the inset map, where the two fan surfaces (T1 - upper; and T2 - lower) are shaded in pink. The graphed elevation data are derived from the 5 m Hellenic Cadastre SA, as is the background hillshade of the map.

**4.** Unit AD in the current manuscript comprises aeolian silty sand and includes terrestrial gastropod shells. The authors argue that the deposition of this unit post-dated abandonment of the T1 surface and this is entirely reasonable. But to assign a depositional age for this unit to the period of incision of T1 gravels (lines 271-274), only because similar aeolian deposits are present around Crete (unreferenced statement), and without proving that they were indeed deposited during this incision phase and prior to deposition of the lower fan gravels, is inappropriate. So dating the gastropod from these aeolian deposits proves little other than that some aeolian silty sand was deposited locally in the late Holocene, necessarily after abandonment of the T1 surface.

**5.** This brings us to the authors' preference, in this instance, to believe radiocarbon ages instead of IRSL ages. The authors state that they collected most of the bulk sediment samples from close to terrace surfaces where the materials

were accessible. As acknowledged within the text, they all have very low total organic carbon contents, but the origin of the carbon within the samples receives little discussion (Section 4.4) regarding whether it is possible that there may have been contamination from plants (living and dead, surface litter and root systems). These contaminants arguably have the potential for minimizing resulting ages, and even making the ages irrelevant to the timing of events they are designed to investigate. The question-marks regarding the radiocarbon ages presented are at least as compelling as the arguments they use to dismiss the validity of our substantially older IRSL ages. Interestingly, the authors do argue for younger contaminants in their landslide deposits to explain their younger ages (lines 399-400).

In lines 395-396 the authors state that "The deposition order obtained from the radiocarbon dating agrees with the sequence of events established in the field." This statement is demonstrably incorrect, as further explored in their following sentences (396-404). Notably, the radiocarbon age for L1 is younger than those for T1 and T2, but the authors claim stratigraphic evidence that L1 pre-dates T1 and T2. By their own pen, the statement is clearly incorrect and should be removed from the manuscript.

**6.** Local soil development is highly variable and is influenced by a number of factors, including climate, parent material (including chemistry) and topography (Lin 2011). Thus, comparing soil development in Klados with areas such as Tsoutsouros in central southern Crete (130 km away) is risky. The Bt and Bk horizons in Tsoutsouros alluvial fans (Gallen et al. 2014) are about 2 m deep and similar horizons at Sfakia (20 km away) range from 5-16 cm (Pope et al. 2008; p 214, Section 7). A B horizon is present on the T1 fan surface at Klados but is limited in depth (Mouslopoulou et al. 2017).

**7.** The manuscript interprets the presence of the double coastal sea cliff at Klados to result from deposition of a landslide and uplift associated with the AD365 earthquake. However, double (or even multiple) sea cliffs are present at different elevations in other coastal fan deposits along southern Crete that lack a landslide source for sediment supply. For example, west of Aradaina Gorge (Figure 2) these sea-trimmed fans are present along a 3 km length of the coastline.

[Figure]

Scale bar 3 km long

**Figure 2:** Double sea-trimmed fans between Agia Roumeli and Aradaina Gorge, southwest Crete.

[Figure]

Figure 3

Similar twin sea cliffs, but at a higher elevation, are present at the settlement of Agia Roumeli, at the mouth of the Samaria Gorge (see Figure 3). Thus, the deposits/processes at Klados/Domata may not be as unique for Crete as the authors present (lines 106, 426, 429 and 503).

In summary, we are pleased that this paper provides new information on the likely presence of a landslide in the upper Klados catchment. The presence of this landslide and its deposits certainly raises the question whether stochastic events may account for geomorphology, erosion and deposition. However, due to the ambiguities associated with inconclusive stratigraphic and geochronological data identified above, this manuscript fails to prove its hypothesis that 'the entire fan and terrace sequence' (lines 22-24) at Klados is late Holocene in age. Thus, in this comment we question some of Bruni et al's primary conclusions, despite the fact that they are presented with such certainty.

*Mouslopoulou, V.**,** Begg, J., Fülling, A., Moraetis, D., Partsinevelos, P., and Oncken, O., 2017. Distinct phases of eustatic and tectonic forcing for late Quaternary landscape evolution soutwest Crete, Greece.* Earth Surface Dynamics *5, 1–17, https://doi.org/10.5194/esurf-5-511-2017, 2017.*

*Lin 2011, Three Principles of Soil Change and Pedogenesis in Time and Space. SSSAJ: Volume 75: Number 6.*

---

## Author Comment (AC2)

**Line-by-line responses to Anonymous Referee #2**

1. Orienting the reader to keep track of all the methodological moving parts is a significant challenge. The manuscript could be **substantially strengthened by (1) further explaining some of the key observations, and (2) reorganizing the text to more consistently separate the results from the discussion.**

If comment (2) refers to the modelling, we see it as a point of discussion. We put the landslide modeling component in the discussion because it is an interpretation of the more substantiated results we obtained from mapping and geochronology. We use this modelling to reinforce the argument of the catastrophic sedimentary input and do not consider it a primary result but supplementary to our interpretation. Because it is an interpretation, positioning it earlier on in the manuscript might be perceived as inappropriate and out of place. However, we understand the reviewer's concern and have worked to streamline the presentation to ease readability.

2. Regarding #1, The Introduction situates the work in the context of strath and fill terraces and alluvial fans. However, the largest geomorphic feature in this study sits squarely on a shoreline, and likely better described as a fluvial fan delta (see Sun et al. (2002), *WRR*, doi: 10.1029/2001WR000284). **How, if at all, does this distinct geomorphic context affect how the present results are related to previous studies for river terraces and alluvial fans in non-coastal settings?** The line-by-line comments below also note several places where the **stratigraphic observations could be more fully explained** (see comments for L220, L238, L311, and L412).

The reviewer brings up a good point about precise terminology and we have revised the manuscript accordingly to describe the coastal fans as "alluvial fan deltas". We used "alluvial fan" in the original submission for consistency with other studies conducted on coastal alluvial fans in Crete and the fact that the stratigraphy preserved in the deposit is not deltaic in nature (e.g. no forests or bottom sets were observed). For clarification, we have also added stratigraphic sections to the manuscript (Fig. 6).

We do not think that the geomorphic context near a coastline affects how our findings relate to previous studies in non-coastal settings. Beyond coastal erosion, the deposits do not bear evidence of strong interactions with sea level or coastal waters (e.g. no topset-foreset pairs). Moreover, the clear continuity between the individual fans and terraces indicates a regular deposition process. This suggests our observations are upstream of significant sea level influence and, therefore, would be largely comparable with alluvial fan and terrace deposits observed in other settings.

3. Regarding #2, I found the text regarding the landslide modeling difficult to follow (see comments for L178, L186, L463, and L454). The **model description appears abruptly in the Introduction**, and could use further description there. Then the **model results are shown in the Discussion** (section 5) rather than the main results section (section 4). As a result, the landslide modeling feels pasted on, rather than integrated with the rest of the work. I think it is an impressive part of the paper, and worthy of inclusion in the formal results.

Indeed, as even a short introduction to the modelling methodology requires a lot of specifics, we decided to include a detailed description in the supplementary section of the manuscript. However, the comment on a more in-depth description of the model in the Introduction is noted, and will be implemented into the revised manuscript. Specifically, we have worked to streamline the writing to improve readability and flow.

As noted above, the modelling is used to reinforce the hypothesis that a landslide caused the aggradation and incision cycles which are at odds with the deposits in the nearby valleys. We arrive at this hypothesis based on our primary field observations and data; it is, therefore, regarded as an interpretation of the result. For this reason, we think it is more appropriate to place all discussion of the landslide modeling in the discussion section of the manuscript. But we are thankful for the comment, and will have to discuss the implications of including it as a formal result.

**Line by line responses to Anonymous referee # 2**

1. L137: "tidal notch" – consider providing a concise definition (and perhaps a citation) for this geomorphic indicator, which seems to be important for this study. Also, it could be helpful to briefly describe how this feature will be "used as a relative age marker" at this point in the text.

This is an excellent point. We have revised the text to: "These paleoshorelines delineate the temporal position of sea level through tidal or bioerosional notches, cemented beachrock, topographic benches, and shore platforms (Chappell, 2009). The uplift of a Holocene paleoshoreline by as much as 9 m a.s.l. on the southwestern coast of Crete is often attributed to an unusually large earthquake (MW 8.3–8.5) in AD 365 (Mouslopoulou et al., 2015; Shaw et al., 2008), but a more recent study suggests that uplift occurred through a series of earthquakes with $Mw < 7.9$ in the first centuries AD (Ott et al., 2021). Regardless of conflicting interpretations, this prominent paleoshoreline is observable along > 200 km of coastline in western Crete and provides a robust Late Holocene time marker. Following Ott et al. (2021), we refer to this Late Holocene coastal feature as the Krios paleoshoreline, based on its maximum elevation at Cape Krios in southwestern Crete." (line 150-158)

2. L164: "Bulk sediment measurements" seems to be a vague title for this subsection, which focuses on radiocarbon dating. Suggest renaming to emphasize dating.

We agree and have clarified this term as "bulk sediment dating".

3. L178: The landslide model appears rather abruptly, and the specific objectives of the modeling are not stated until the end of this section (L196-200). For clarity, consider moving these objects to the start of the section. More explanation is also needed for these rheology models (e.g., Voellmy – not familiar with this model).

We agree with the reviewer and will introduce the aims of the modelling and the rheology models more clearly, possibly along the following lines: "To test the feasibility of the hypothesis that a rockfall turned landslide provided the necessary material to form the large sedimentary deposits throughout the valley, we utilised [...]" (213-214).

"Several studies report successful model results for landslides when a Voellmy or frictional rheology is used as the basal rheology, and several back-analysed historical events are available using these rheologies  (Aaron and Hungr, 2016; Grämiger et al., 2016; Hungr, 1995; Nagelisen et al., 2015). Adding to the basic frictional rheology equation, Voellmy rheology includes a "turbulent term" which is dependent on flow velocity and the density of the material and summarises the velocity-dependent factors of flow resistance (Hungr and Evans, 1996)." (line 216-220)

4.  L186 "pre-landslide topography" – clarify whether you reconstructed the pre-failure surface for the landslides source area.

We revised the text here for clarity as suggested. We also point the reader to section 4.6. Volumes of rockfall and valley infill (line 411-417).

We quote from the revised text: "We produced a DEM of the modern landscape without the Holocene deposits mapped in this study as the pre-landslide topography (DEMpre). For this, the thicknesses of all deposits were subtracted from the present-day topography (Fig. S2). The pre-failure surface for the source area was reconstructed using the thicknesses of the reconstructed rockfall wedges creating a rough minimum estimate of the mountain face's bedrock topography before the landslide event." (line 226-230)

5.  L211: Figure 3: for clarity, assign the sketch in the upper left as a formal subfigure (subfigure ("a"). Suggest also adding a word or two to describe each of T1, T2, T3, and L1. Nice use of human for scale!

Good point. We have revised the figure accordingly.

6.  L220: "that T2 unconformably overlies a paleo-beach deposit" – this seems like one of the key observations to establish a new chronology for this landscape (and is highlighted in the abstract). Yet the observation goes by quickly and is tucked away (Fig. 4e) in part of a very busy figure. I suggest expanding this description, particularly to build the case that this is a paleo-beach deposit. Some of the related text comes in L263-264, but presenting all of the observations together would make it easier to follow.

This is a good point. We wanted to separate the results and interpretation strongly in the original submission, but recognize that this is a critical observation. We have therefore revised the text to add more discussion of this key finding here.

We have also added a new figure highlighting this key observation.

[Figure]

**Figure 5:** The contacts between the tidal notch, $T_2$, and the paleobeach are illustrated by photographs from the west side of the study area. (a) Overview showing the unconformable relationship of the Late Holocene tidal notch and the $T_2$ fan highlighting the location of figures in other panels. (b) Oblique aerial perspective view of the outcrop with the major features highlighted. (c) Detail of the Vermetid extraction site shows how gravels of $T_2$ overlie a Vermetid shell pocket in the tidal notch. (d) Detail of the contact zone between the carbonaceous bedrock, $T_2$, and the tidal notch (partly buried by colluvium). (e) The Vermetid fossil pocket is covered by $T_2$ fan material (detail of (c)).

7. L238: the subfigures in Figure 4 are discussed out of sequence, which makes the argument more difficult to follow.

We thank the reviewer for this comment. We corrected the sequence to follow the appearance in text in the revised manuscript.

8. Throughout: "Aeolian" → "aeolian" or "eolian"

We use "eolian" in the revised manuscript.

9. L296: "river attempted to adjust its slope" – be careful about anthropomorphizing (a river cannot attempt to do anything).

Fair point. We revised this sentence.

10. L297-298: "deposits change vertically from unsorted debris flows at the bottom to layered sheet flows" – correct usage is "debris flow deposits" and "sheet flow deposits."

We made this change.

11. L311-312: The observed radiocarbon ages from the shells – 800 to 1000 years older than the inferred age of the uplift that raised the notch above sea-level – seems to pose a significant complication for the proposed timeline of events. For this scenario to hold, the shells would have needed to have been preserved for 800 years after the organisms' death. Is that plausible? This issue goes beyond my expertise, but I am curious. Perhaps an additional sentence or two, or a related example from the literature, could flesh out this point.

Firstly, the reported radiocarbon ages cannot be directly compared with calendar years, as they have not been calibrated. We adjusted the manuscript to include calibrated calendar years of the fossil dates to ease comparison, which reduces the discrepancy. Secondly, there are three options to explain the old ages. Either (1) the paleoshoreline (tidal notch) was not uplifted in one single event as proposed in previous literature (Pirazzoli et al., 1982, 1996; Shaw et al., 2008; Stiros, 2001), but is the result of gradual uplift (Ott et al., 2021), or (2) the organisms were killed and preserved by intermittent burial by older $T_1$ deposits, or (3) the organisms have really been preserved for this amount of time. We lack data to distinguish between these possibilities but none of these options has any effect on our primary conclusions.

12. L356: In Table 2, it is unclear why there are 4 numbers listed under "Intermediate." The text mentions 6 wedges, is that related?

Thank you for the comment, we will clarify in the text that of the 6 wedges, 2 relate to the maximum and minimum values and only 4 to the intermediate-sized wedges. It is worth highlighting that the maximum value is oversized and was not used in any of the subsequent analyses.

13. L412-413: The comparison of the radiocarbon dates with the existing IRSL dates is a critical point in this paper. I suggest going a bit further to explain why you think the IRSL dates could be biased, particularly in a way that is accessible to those outside the geochronology community. You think the IRSL samples included "of a mix of bleached and unbleached grains resulting in late Pleistocene ages" – can you expand on this point using more accessible language?

We thank the reviewer for this comment. We revised the text to provide a more detailed description of the biases that the previously published IRSL samples might suffer from. We quote from the revised text: "Luminescence burial dating of deposits exploits the assumption that charge is gradually built up in feldspar or quartz grains due to radiation from radiogenic

decay of radioactive elements and cosmic rays. To relate the amount of charge a grain releases as luminescence signal to the duration of sediment burial (depositional time of unit), all charge within the crystal lattice needs to be fully released by sun bleaching before deposition; a process that requires seconds of full sun exposure for quartz and minutes for feldspar (Rhodes, 2011). Alluvial fans, especially in small catchments with short transport and a significant portion of debris flow deposits, are therefore prone to biases in luminescence measurements because the short transport in sediment-rich flows usually does not allow for a complete bleaching of the mineral grains, and especially not feldspar (Rhodes, 2011). This effect is enhanced because minerals freshly released from the bedrock have worse luminescence characteristics and take longer to bleach (Rhodes, 2011).

The anomalously old luminescence ages reported by Mouslopoulou et al. (2017) are likely biased due to incomplete bleaching caused by the turbulent mode of transport (Rhodes, 2011). The broad positively skewed age distributions of measured equivalent dose measurements (the amount of charge released from the grains) in Mouslopoulou et al. (2017) from feldspar IRSL indicate a mix of bleached and unbleached grains resulting in late Pleistocene ages for both fan units. The mixture of bleached and unbleached grains is especially evident because Mouslopoulou et al. (2017) also measured the quartz OSL signal, and found the same positively skewed age distributions but with younger ages. The discrepancy between the younger quartz OSL and older feldspar IRSL measurements can be explained by the more rapid bleaching of quartz grains; however, these authors discarded and did not report the OSL ages choosing instead to construct their interpretation on the IRSL measurements alone." (line 478-498)

14. L463-464: How was the "best fit" model determined?

We added some text to this point in the revision. In short, we largely relied on runout distance, speed and model thickness to define the best-fitting model. For example, we discarded models with maximum slide velocities of sound speed or larger, and travel times of less than 1 minute (see Table 3). The best-fit model reproduces our field observations of deposits up to 100 m above the modern stream channel, and reports the most realistic natural outflow, but of course still contains a lot of assumptions.

15. L454-501: Section 5 is the Discussion, but these lines present a lot of additional results. Consider moving this material earlier in the manuscript.

The reviewer raises an important point that we discussed during the process of writing this manuscript. Though the landslide modelling does show important additional results that are presented in the discussion, the whole idea of doing a landslide runout model hinges on the interpretation of the alluvial deposits. To generate a logical flow and now jump ahead with interpretations in the result section, we chose to present these results in the discussion section of the manuscript.

16. L511-536: Can you tie this sequence to Figure 8 using specific references to each of the subfigures?

Yes, we can (Sect. 5.5; Fig. 10).

**References used by the authors in the response**

Aaron, J. and Hungr, O.: Dynamic analysis of an extraordinarily mobile rock avalanche in the Northwest Territories, Canada, Can. Geotech. J., 53(6), 899–908, doi:10.1139/cgj-2015-0371, 2016.

Chappell, J. M.: Sea level change, quaternary, in Encyclopedia of Earth Sciences Series, pp. 658–662, Springer Netherlands., 2009.

Grämiger, L. M., Moore, J. R., Vockenhuber, C., Aaron, J., Hajdas, I. and Ivy-Ochs, S.: Two early Holocene rock avalanches in the Bernese Alps (Rinderhorn, Switzerland), Geomorphology, 268, 207–221, doi:10.1016/j.geomorph.2016.06.008, 2016.

Hungr, O.: A model for the runout analysis of rapid flow slides, debris flows, and avalanches, Can. Geotech. J., 32(4), 610–623, doi:10.1139/t95-063, 1995.

Hungr, O. and Evans, S. G.: Rock avalanche runout prediction using a dynamic model, Proc. 7th Int. Symp. Landslides, Trondheim, Norw., 17, 21 [online] Available from: http://www.clara-w.com/DANWReference2.pdf, 1996.

Mouslopoulou, V., Nicol, A., Begg, J., Oncken, O. and Moreno, M.: Clusters of megaearthquakes on upper plate faults control the Eastern Mediterranean hazard, Geophys. Res. Lett., 42(23), 10282–10289, doi:10.1002/2015GL066371, 2015.

Mouslopoulou, V., Begg, J., Fülling, A., Moraetis, D. and Partsinevelos, P.: Distinct phases of eustatic and tectonic forcing for late Quaternary landscape evolution in southwest Crete , Greece, Earth Surf. Dyn., 5, 511–527, 2017.

Nagelisen, J., Moore, J. R., Vockenhuber, C. and Ivy-Ochs, S.: Post-glacial rock avalanches in the Obersee Valley, Glarner Alps, Switzerland, Geomorphology, 238, 94–111, doi:10.1016/j.geomorph.2015.02.031, 2015.

Ott, R. F., Wegmann, K. W., Gallen, S. F., Pazzaglia, F. J., Brandon, M. T., Ueda, K. and Fassoulas, C.: Reassessing Eastern Mediterranean tectonics and earthquake hazard from the AD 365 earthquake, AGU Adv., doi:10.31223/X5H036, 2021.

Pirazzoli, P. A., Thommeret, J., Laborel, J. and Montaggioni, L. F.: Crustal Block Movements from Holocene Shorelines: Crete and Antikythira (Greece), Tectonophysics, 86, 27–43, 1982.

Pirazzoli, P. A., Laborel, J. and Stiros, S. C.: Coastal indicators of rapid uplift and subsidence: examples from Crete and other eastern Mediterranean sites, Zeitschrift Fur Geomorphol. Suppl., 102(1996), 21–35 [online] Available from: http://www.scopus.com/inward/record.url?eid=2-s2.0-0029732821%7B&%7DpartnerID=40%7B&%7Dmd5=4b91f23e3f100447fd0a5686efeb29da , 1996.

Rhodes, E. J.: Optically Stimulated Luminescence Dating of Sediments over the Past 200,000 Years, Annu. Rev. Earth Planet. Sci., 39(1), 461–488, doi:10.1146/annurev-earth-040610-133425, 2011.

Shaw, B., Ambraseys, N. N., England, P. C., Floyd, M. A., Gorman, G. J., Higham, T. F. G., Jackson, J. A., Nocquet, J.-M., Pain, C. C. and Piggott, M. D.: Eastern Mediterranean tectonics and tsunami hazard inferred from the AD 365 earthquake, Nat. Geosci., 1(4), 268–276, doi:10.1038/ngeo151, 2008.

Stiros, S. C.: The AD 365 Crete earthquake and possible seismic clustering during the fourth to sixth centuries AD in the Eastern Mediterranean: A review of historical and archaeological data, J. Struct. Geol., 23(2–3), 545–562, doi:10.1016/S0191-8141(00)00118-8, 2001.

---

## Author Comment (AC3)

**Response to J. Begg, V. Mouslopoulou, D. Moraetis**

**COMMENTS ON Bruni et al. 2021**

J. Begg, V. Mouslopoulou, D. Moraetis

6 April 2021

We are the principal authors of Mouslopoulou et al. (2017), the conclusions of which are challenged by this submission.

The Domata/Klados River area is a beautiful and under-appreciated area of Crete and this manuscript by Bruni et al. discusses the relationship between a large landslide event and deposition within a confined catchment on the southern side of the island. We believe that while the significance of the landslide event in the headwaters of the Klados River is credible, as are some of the deductions that they have made regarding its impact on deposition through the catchment, there are important elements within this manuscript that are not as straightforward as the authors have presented. We will explore some of these issues in the comments below.

**1.** The authors claim that the alluvial deposits beneath the surface T2 post-date a regional-scale earthquake in AD365 that uplifted the coastline at Klados by c. 6 m. If this is true, most of the conclusions of this work are correct. If not, however, many of their conclusions are demonstrably wrong. Thus, the authors, in our view, should have taken special care to demonstrate solidly this relationship. Below we show that they have not.

We thank the reviewers for this comment. As pointed out in a reply to Mouslopoulou et al. (2017) by two of the authors of this manuscript (Gallen and Wegmann), what we call the $T_2$ fan forms a buttress unconformity with the late Holocene erosional notch. This was clearly shown in figures in that 2017-comment and is shown again in this manuscript, along with additional supporting information. This observation demands that deposition of the $T_2$ fan post-dates uplift of the late Holocene notch. This primary observation of a simple cross-cutting relationship is not in question and is definitive evidence that the $T_2$ fan is late Holocene in age. This relationship is even shown in Figure 6b of Mouslopoulou et al. (2017).

We added new photos and enlarged photos to more clearly show the cross-cutting relationship to the revised manuscript. We thank the reviewer for this request as it strengthens the presentation of our study.

The relationship between the alluvial deposits underlying surface T2 and the AD365 "tidal notch" is not clearly presented. The authors in lines 233-234, 289-290 (and elsewhere) repeatedly claim that the AD365 "tidal notch" is overlain by alluvial deposits underlying terrace T2. However, neither Figure 4h nor 4i show this. Instead, these figures show the 365 AD tidal notch preserved on limestone bedrock (Fig. 4h) but missing from nearby gravels (Fig. 4h and 4i).

As noted above, we have revised the presentation of the basic field observations (see Fig. 5). However, the reviewers make a confusing comment here about the notch missing from the gravels. Yes, the notch is in the limestone bedrock and continues behind the $T_2$ alluvial gravels

(this is a buttress unconformity); this is the entire point of showing the figure.

We cannot be certain, but the reviewers seem to imply that the lack of preservation of a tidal erosion notch in the fan is somehow damaging to our arguments, which is entirely wrong. The notch is not observed in the gravels because the fan is younger than the notch. Also, the fan is highly erodible and unlikely to preserve a notch even if it did exist. Hence our confusion.

Alternatively, perhaps what the reviewers meant by this comment was that the notch formed at the front of the fan deposits coeval with its formation across the limestone headlands on either side of the Klados Gorge, and now is eroded away due to back-wasting of the $T_2$ alluvial gravel deposits by wave and gravitational action. This is a possible scenario if $T_2$ fan formation was Pleistocene; however, we show through stratigraphic observations (e.g., the existence of a buttress unconformity between the limestone headland that includes the late Holocene notch with Vermetid gastropod encrustations) and the existence of a late Holocene paleo beach deposit that is buried by the younger $T_2$ alluvial deposits, that this hypothesis is not supported by available stratigraphic information. We have included a new figure that clearly shows these observations.

[Figure]

**Figure 5:** The contacts between the tidal notch, $T_2$, and the paleobeach are illustrated by photographs from the west side of the study area. (a) Overview showing the unconformable relationship of the Late Holocene tidal notch and the $T_2$ fan highlighting the location of figures in other panels. (b) Oblique aerial perspective view of the outcrop with the

major features highlighted. (c) Detail of the Vermetid extraction site shows how gravels of $T_2$ overlie a Vermetid shell pocket in the tidal notch. (d) Detail of the contact zone between the carbonaceous bedrock, $T_2$, and the tidal notch (partly buried by colluvium). (e) The Vermetid fossil pocket is covered by $T_2$ fan material (detail of (c)).

The "tidal notch" is not a deposit, it is a geomorphological feature, the result of local modification of the bedrock, here limestone, by marginal marine processes. The limestone is well lithified while the alluvial gravels are "unconsolidated" (see Section 4.3) and both lie at the inland extent of today's active beach. There is no discussion of the potential for these active marginal marine processes to erode these two lithologies differently. Would the AD365 "tidal notch", even if it had been present on the alluvial gravels (should they really be older), have been preserved? Why do they authors fail to consider this alternative scenario? The images presented do not identify the contact between limestone bedrock and "T2 deposits" (and therefore the relationship). Further, the cliff on the right-hand side of Fig. 4i comprises T2 alluvial materials and doesn't show the "tidal notch", but that does not mean that it wasn't once there before erosion by active marginal marine processes. This point is critical to the arguments that "T2 infill deposits" (all 20 m of them) post-date the AD365 uplift event that is asserted in the rest of the paper.

We are aware that the notch is not a depositional feature and we do not state otherwise in the manuscript. We also recognize that the erodibility of the bedrock limestone and fan deposits are different. However, the fact that the $T_2$ fan covers the notch indicates that the $T_2$ deposit is younger than the notch. This is a basic cross-cutting relationship regardless of differences in erodibility and "marine trimming". We note that we are not the first scientists to make this basic observation. Booth (2010) conducted a detailed study of several coastal catchments in southern Crete with a particular emphasis on the Klados catchment. In this study, they independently report the same observation; the $T_2$ fan covers the notch, thus this buttress unconformity demands that the deposition of the $T_2$ fan postdates the late Holocene uplift of this paleoshoreline.

As noted above, we made revisions to the presentation of the figures to better illustrate this cross-cutting relationship.

[Figure]

[Figure]

Further, if the authors' interpretation above is correct: 1) the deposition of the "T2 infill deposits", 2) erosion of the lower coastal cliff and 3) incision by the Klados River below the T2 surface is required to have occurred after the AD365 earthquake. In such a scenario, the speed of deposition of the "T2 infill deposits" and their incision (by sea and river) to their present day configurations must have been exceptionally fast, with only 1600 years available to complete. Given the small catchment area and limited water flow, these events are less likely.

Yes, this is the entire point of the study and why it is so interesting. Considering the observation of the valley filling landslide deposit that is highly erodible (a critical observation missed in (Mouslopoulou et al., 2017), this scenario is credible, likely and indeed demanded by basic cross-cutting stratigraphic field relationships. It shows how such a small catchment in the aftermath of a large sediment pulse can become ultra-sensitive to external perturbations, e.g. storms and earthquake sediment mobilization that rapidly aggrade and incise the deposits. **The significance of this study is to show that thick sequences of alluvial deposits can form in a very limited time in the aftermath of a sediment pulse, contrary to the traditional interpretation of tectonic and climatic forcing.**

It is worth highlighting that there is a growing body of literature that shows these "stochastic" events and associated rapid development of thick alluvial deposits are more common than previously recognized and have often been inappropriately interpreted as the result of long-term climate change. We invite the comment authors to read Scherler et al. (2016), which shows how a sequence of river terraces traditionally assumed to be linked to the early to mid-Pleistocene variations in climate, turned out to be due to a Holocene landslide. The study shows aggradation and incision of a similar number of terraces with similar terrace thicknesses within the Holocene in the semi-arid landscape of California. We also point the reviewers to the excellent work of Schwanghart et al. (2016) and Stolle et al. (2017) that show large alluvial infill deposits in the Central Himalaya in Nepal are Holocene in age and related to large-scale landslides up-valley.

**2.** The unexplored problems associated with the "tidal notch" and deposition of the "T2 infill deposits" discussed above, are compounded by using their interpreted relationship to assume that the "paleobeach" deposit underlying "T2 infill deposits" must represent the AD365 shoreline. This is unproven. Instead, this correlation is based on the relationship that we questioned in (1) and on the elevation of each of the features. We argue that in our model (see Mouslopoulou et al., 2017) we would expect a "paleobeach" deposit seaward of the base of the marine cliff that truncates T1 – thus, this observation does not contradict an older age of the alluvial fans.

This comment is moot based on the cross-cutting relationships observed between the $T_2$ fan and notch described in detail above, so we have not made any revisions to address it. Nonetheless, we use this as an opportunity to highlight issues with the interpretations presented in Mouslopoulou et al. (2017) and detail why our inference that this paleobeach represents the 365 AD shoreline is more reasonable given the data.

The stratigraphic observation of the $T_2$ terrace overlying a paleobeach deposit was missed in the original submission by Mouslopoulou et al. (2017), but highlighted in the comment by Gallen and Wegmann (2017). In revision of their manuscript, Mouslopoulou et al. (2017) included mention of this paleobeach and suggested that it is Pleistocene in age. However, the authors did not consider the fact that the paleobeach deposit is found at the same elevation as the erosional notch, which would be a remarkable coincidence if it were Pleistocene. If this paleobeach were Pleistocene, the traces of the late Holocene shoreline that is found on both sides of the modern Domata beach would have been completely eroded away in the center of the modern bay with erosion revealing an older Pleistocene paleobeach that is found at the exact same height as the Late Holocene one. Additionally, considering that the $T_2$ fan covers (buries) both the notch and the beach, it is reasonable and more parsimonious to assume that the notch and paleobeach indeed represent the same paleoshoreline.

During field work we have taken a luminescence sample from the paleobeach. In contrast, to the fan deposits luminescence dating of beach deposits is more promising, because the constant swash of beach material provides better conditions for grain bleaching. However, given the clear field relationship, and the negligible quartz and feldspar content of local rocks, we decided there is no benefit nor need to date this sample. If the reviewers still have any doubt about the Holocene age of this paleobeach after our presentation of additional field pictures with clear cross-cutting relationships, they are welcome to process this sample.

**3.** Reference is made by the authors to the "crisp" similarity in morphology of the two marine cliffs at Klados mouth. More careful examination of this statement shows that this is not true. The 5 m topo DEM that the authors used to derive their data is entirely adequate to contradict this assertion. See below profiles 1 to 3 across the Klados beach that illustrate that the lower sea-cliff is significantly steeper than the upper sea-cliff (75° vs. 53° average slopes). In addition, the base/crest of the lower-cliff is much sharper than those of the upper-cliff. The morphological differences between the two sea-cliffs are indicative of an age difference substantially more than 1600 years. These observations undermine the authors' assertion that the morphologies are equally immature and therefore both of late Holocene age and provide critical corroborative evidence that the upper sea-cliff is substantially older than the lower sea-cliff.

[Figure]

Figure 1 Elevation profiles at Klados River mouth, illustrating the contrasting morphologies of the upper and lower seacliffs. Profile lines are in black, but seacliffs are colour-coded, dark blue for the upper and red for the lower (note that the vertical scale = the horizontal scale). Profile locations are shown on the inset map, where the two fan surfaces (T1 - upper; and T2 - lower) are shaded in pink. The graphed elevation data are derived from the 5 m Hellenic Cadastre SA, as is the background hillshade of the map.

This is an intriguing comment that is very similar to a comment made by Gallen and Wegmann (2017) regarding issues of the interpretations presented in Mouslopoulou et al. (2017). We thank the reviewers for producing this figure, which provides an opportunity to highlight why our interpretations are more favorable than those presented in Mouslopoulou et al. (2017).

First, the figure above selectively chooses the steepest profile of $T_2$ (in red above, profile 3) to argue that the $T_2$ sea cliff is steeper. The *__active__* sea cliff for $T_2$ Profiles 1 and 2 have slopes of ~55-60 degree, which is remarkably similar to the slope of the $T_1$ paleo-sea cliff, supporting our statements in the manuscript that these erosional cliffs are "similarly crisp". Also see the detailed topographic profile in Figure 5a in Mouslopoulou et al. (2017) for evidence of the similar morphology of these two sea cliffs.

Second, the $T_2$ sea cliff is actively eroding by wave action during winter storms. As such, we expect that some portions will be oversteepened (profile 3), so this observation is not damaging to our interpretations.

Third, as pointed out in Gallen and Wegmann's (2017) comment , the similar sharpness of the $T_1$ and $T_2$ sea cliffs as shown in the figure above is very problematic for the interpretations presented in Mouslopoulou et al. (2017). Both terraces consist of largely uncemented and unconsolidated granular material. In our interpretation, the $T_1$ sea cliff is only ~1600 years old, which explains why it maintains a steep angle of 53 degrees, similar to the actively eroding $T_2$ sea cliff. The

interpretation of Mouslopoulou et al. (2017) suggests the $T_1$ sea cliff is >30 kyrs older than the active $T_2$ sea cliff. Considering that these are unconsolidated granular deposits, how does $T_1$ maintain such sharpness over that duration of time? This presents a problem for the interpretations presented in Mouslopoulou et al. (2017), but is easily explained by our preferred interpretation.

In response to this comment, we produce our own profiles to show the similar sharpness of the sea cliff and elaborate on why this supports the interpretations that $T_1$ is young (e.g. Holocene). We thank the reviewers for pushing us to more strongly support our interpretations with quantitative analysis of the sea cliff morphology (see supplementary sect. 7, Fig. S6). We are confident that this will be helpful in convincing the reader of their young age.

[Figure]

**Figure S6**: Structure from motion (SfM) photomosaic, digital surface model (DSM), and diffusion modelling results. (a) and (b) show the SfM photomosaic and DSM result, respectively, along with the location of the two swath profiles (LF – lower fan, UF – upper fan). (c) and (d) are the minimum elevations of the swath profile (grey lines), which are assumed to approximate the vegetation-free fan morphology. Also shown on both plots is the initial (blue line) and the final modelled (green line) topographic profiles for the upper fan. The bold grey line shows the data used in the

diffusion modelling. (e) and (f) show the best fit diffusion coefficient, D, results for the Holocene and Pleistocene age models as the white and great vertical rectangles, respectively, plotted against mean annual precipitation (MAP). Also shown is the global compilation of diffusion coefficients from Richardson et al. (2019) classified based on substrate (e) and overlying vegetation (f).

Unit AD in the current manuscript comprises aeolian silty sand and includes terrestrial gastropod shells. The authors argue that the deposition of this unit post-dated abandonment of the T1 surface and this is entirely reasonable. But to assign a depositional age for this unit to the period of incision of T1 gravels (lines 271-274), only because similar aeolian deposits are present around Crete (unreferenced statement), and without proving that they were indeed deposited during this incision phase and prior to deposition of the lower fan gravels, is inappropriate. So dating the gastropod from these aeolian deposits proves little other than that some aeolian silty sand was deposited locally in the late Holocene, necessarily after abandonment of the T1 surface.

This is a fair point and we have made revisions qualifying the results. However, we note that even without this geochronology, the cross-cutting, stratigraphic and geomorphic observables support our interpretation that the deposits in Klados are Holocene and not Pleistocene.

**5.** This brings us to the authors' preference, in this instance, to believe radiocarbon ages instead of IRSL ages. The authors state that they collected most of the bulk sediment samples from close to terrace surfaces where the materials were accessible. As acknowledged within the text, they all have very low total organic carbon contents, but the origin of the carbon within the samples receives little discussion (Section 4.4) regarding whether it is possible that there may have been contamination from plants (living and dead, surface litter and root systems). These contaminants arguably have the potential for minimizing resulting ages, and even making the ages irrelevant to the timing of events they are designed to investigate. The question-marks regarding the radiocarbon ages presented are at least as compelling as the arguments they use to dismiss the validity of our substantially older IRSL ages. Interestingly, the authors do argue for younger contaminants in their landslide deposits to explain their younger ages (lines 399-400).

The field relationships clearly show that the IRSL results of Mouslopoulou et al. (2017) are unreliable. The $T_2$ fan forms a buttress unconformity with the Holocene notch, requiring its deposition in the Late Holocene. The IRSL results of this Holocene deposit produce an apparent age of ~40 kyr with significant scatter in the equivalent doses indicating the results are incorrect. Furthermore, we remind the reviewers that their IRSL results are not stratigraphically consistent; they suggest that the relatively younger $T_2$ fan was deposited BEFORE the older $T_1$ fan.

The unreliability of the IRSL-feldspar results is not surprising. The Klados catchment is small and the deposits are high-energy and close to the source with a significant amount of debris flow deposits. This is problematic for luminescence dating (and especially feldspar IRSL) because the environmental conditions are poor and incomplete bleaching before deposition is likely. This was a point raised by Gallen and Wegmann (2017) regarding the IRSL results present in Mouslopoulou et al. (2017), which the authors never adequately addressed.

The distributions of paleodoses from IRSL measurements support the notion that the results suffer

from incomplete bleaching since equivalent doses are widely scattered and show a skew towards younger ages. Mouslopoulou et al. (2017) acknowledge that their IRSL data suggest incomplete bleaching (quote: "This can indicate insufficient exposure of the sediment to daylight during the last sedimentation cycle."). Perhaps most importantly, Mouslopoulou et al. (2017) state that they generated optically stimulated luminescence (OSL) results for quartz grains. However, they do not report these results, stating "In contrast, the investigated quartz from Domata showed poor luminescence properties: the OSL signals were dim, dose recovery tests yielded unsatisfactory results, the highly scattering palaeo-doses produced positively skewed broad distributions and the resulting quartz ages showed no relationship with stratigraphy (underestimation of true age)." Mouslopoulou et al., (2017) report the same behavior of broad and positively skewed equivalent dose distributions for their OSL measurements as for IRSL measurements. However, they chose to not publish the OSL results stating a "underestimation of true age".

We emphasize that the augments laid out in this quote used to rationalize not reporting the OSL results can equally apply to Mouslopoulou et al.'s (2017) IRSL results. We note that quartz bleaches faster than feldspar, and it is likely that the OSL results are better approximations of the depositional age of these Klados fans and terraces. However, we emphasize that the wide positively skewed scatter in OSL and IRSL measurements clearly points towards incomplete bleaching, which is also acknowledged in Mouslopoulou et al. (2017). The combination of younger OSL ages compared to IRSL, the broad skewed dose distributions for OSL and IRSL, and the poorly suitable depositional environment for luminescence burial dating indicate that the IRSL data reported by Mouslopoulou et al. (2017) are unreliable. We have added two paragraphs to the revised manuscript that discuss the points mentioned above in detail.

We also invite the authors of Mouslopoulou et al. (2017) to read the EGU21 abstract from Schwanghart et al. (Abstract). They perform luminescence dating on alluvial deposits in the Himalaya that are also related to upstream mass movements. They find that despite a transport distance significantly larger than in the Klados catchment, basically no bleaching of feldspar grains occurred during transport in the sediment laden floods and/or debris flows. We assume that the same applies to the feldspar grains measured by Mouslopoulou et al. (2017).

As noted in our manuscript, there is a great deal of uncertainty in our bulk radiocarbon ages. However, they are indeed consistent with the field observations and cross-cutting relationships that require most of the fan and terrace sequence to be Holocene. We consider this secondary evidence in support of the primary stratigraphic and cross-cutting relationships.

The reviewer brings up a good point that we clarify in the revision; the samples collected from $T_2$ and $T_1$ were from recently cut exposures well below the depth of soil, leaf litter, and rooting systems. So these sources of uncertainty for these deposits are small given our sampling approach. The landslide deposit consists of extremely weak material and clean, recent exposures were difficult to access. Because of this, we could not obtain samples from "ideal" locations, and we sampled the best locations possible. As such, it is possible that the samples acquired from the landslide deposit suffer from the sources of uncertainty mentioned above. We include a more

detailed discussion of these points in the revision.

We also want to use this as a chance to highlight that the absolute geochronology for deposits in Klados is a challenge and merits future work. That said, $T_2$ post-dates the Holocene erosional notch, so it is Holocene.

In lines 395-396 the authors state that "The deposition order obtained from the radiocarbon dating agrees with the sequence of events established in the field." This statement is demonstrably incorrect, as further explored in their following sentences (396-404). Notably, the radiocarbon age for L1 is younger than those for T1 and T2, but the authors claim stratigraphic evidence that L1 pre-dates T1 and T2. By their own pen, the statement is clearly incorrect and should be removed from the manuscript.

This is a good point and we will revise the statements accordingly. However, we do discuss in detail on lines 397-400 of the original submission why these discrepancies likely exist. Furthermore, this mismatch highlights the importance of the relative age control that we establish and the cross-cutting relationships observed. These are the primary observations in the study and support our interpretations, the geochronology is supplementary, but helpful. We also note the geochronology of Mouslopoulou et al. (2017) is out of stratigraphic order, and the authors of that study seem comfortable with that when publishing their work.

The modified sentence reads as follows: "Except for one outlier, the deposition order obtained from the radiocarbon dating agrees with the sequence of events established in the field" (Table 1 and line 464-466).

**6.** Local soil development is highly variable and is influenced by a number of factors, including climate, parent material (including chemistry) and topography (Lin 2011). Thus, comparing soil development in Klados with areas such as Tsoutsouros in central southern Crete (130 km away) is risky. The Bt and Bk horizons in Tsoutsouros alluvial fans (Gallen et al. 2014) are about 2 m deep and similar horizons at Sfakia (20 km away) range from 5-16 cm (Pope et al. 2008; p 214, Section 7). A B horizon is present on the T1 fan surface at Klados but is limited in depth (Mouslopoulou et al. 2017).

Based on our own field observation, no B-horizon is present in the $T_1$ fan gravels. This was also evident in photos present in figures 8b and c in Mouslopoulou et al. (2017). The photo is annotated with "possible B horizon", but there is no evidence of clay of calcium carbonate accumulation. The soils, or more accurately, the lack of soil development in the Klados fan deposits supports a young, likely Holocene age and the photos in Mouslopoulou et al. (2017) Figure 8 support the notion that they are immature relative to those developed on Pleistocene fans studied elsewhere on Crete.

We are well aware of the state factors that affect soil development including parent material, climate, topography, drainage, etc. The coastal climate of southern Crete is not substantially different from location to location and the fans described by Pope et al. (2008) and Gallen et al. (2014) are developed on similar alluvial fan material dominated by carbonate grains. However,

the soils are very different in these locations and support our interpretations. A key observation is that many (not all) Pleistocene fans composed of carbonate grains in Crete calcify quickly (see observations presented by the reviewers in the following comment). The Klados fans are almost entirely carbonate and are not cemented at all, in contrast to their Pleistocene counterparts. Indeed the images of the deposits in the following comment below show how different the Pleistocene-age deposits are near Aradena Gorge (~13 km east of Klados) relative to the Holocene features in Klados.

[Figure]

**Figure S5:** Minor soil development on T3 (a), T2 (b), and T1 (c) results in low soil maturity. Typically, a surface horizon of non-degraded organic matter such as pine needles overlies the original alluvial deposits. Soil formation may be accelerated in close proximity to larger plants such as pine trees, but we find no sign of wide-spread pedogenesis. (d) Outcrop "Alta Paleohora" (20 km W of Klados, exact location noted) showing dated MIS 4 alluvial fan material over MIS 5.1 beach deposits (Pope et al., 2008). (e) Outcrop in Paleohora (exact location noted), carbonaceous terrace of MIS 2. B = top soil, C = source rock, K= secondary carbonates, T = clay-enriched (IUSS Working Group WRB, 2015).

**7.** The manuscript interprets the presence of the double coastal sea cliff at Klados to result from deposition of a landslide and uplift associated with the AD365 earthquake. However, double (or even multiple) sea cliffs are present at different elevations in other coastal fan deposits along southern Crete that lack a landslide source for sediment supply. For example, west of Aradaina Gorge (Figure 2) these sea-trimmed fans are present along a 3 km length of the coastline.

[Figure]

Scale bar 3 km long

**Figure 2:** Double sea-trimmed fans between Agia Roumeli and Aradaina Gorge, southwest Crete.

[Figure]

Similar twin sea cliffs, but at a higher elevation, are present at the settlement of Agia Roumeli, at the mouth of the    Samaria Gorge (see Figure 3). Thus, the deposits/processes at Klados/Domata may not be as unique for Crete as the  authors present (lines 106, 426, 429 and 503).

The south coast of Crete comprises marine terrace sequences with numerous paleoshorelines (e.g., Mouslopoulou et al., 2015; Ott et al., 2019). Sequences of sea cliffs are not unique to the Klados/ Domata area and are not presented as such in the manuscript. We therefore did not make any modifications in response to this comment.

We also want to use this as an opportunity to highlight key observations that we made about the uniqueness of the Klados alluvial deposits; although it was not raised by the reviewer. Upvalley from Domata beach within the Klados catchment, the coastal fans are fluvial terraces (they are the same deposits) and extend nearly to the headwaters of the catchment. It is highly unusual in Crete or elsewhere to find an alluvial terrace in drainages this small, suggesting that the conditions in Klados are different than elsewhere. We note that Mouslopoulou et al. (2017) did not report observations of these terraces nor their upstream extent, but these deposits are essential to understanding the origins, history, and deposition of the coastal fans in much the same way the

landslide deposit is critical to understand why this small catchment is capable of generating such large alluvial deposits.

In summary, we are pleased that this paper provides new information on the likely presence of a landslide in the upper Klados catchment. The presence of this landslide and its deposits certainly raises the question whether stochastic events may account for geomorphology, erosion and deposition. However, due to the ambiguities associated with inconclusive stratigraphic and geochronological data identified above, this manuscript fails to prove its hypothesis that 'the entire fan and terrace sequence' (lines 22-24) at Klados is late Holocene in age. Thus, in this comment we question some of Bruni et al's primary conclusions, despite the fact that they are presented with such certainty.

We thank the reviewers for their time in helping clarify points made regarding the Holocene age of the depositional feature in Klados. Their effort has strengthened our arguments. For that we are appreciative.

*Mouslopoulou, V., Begg, J., Fülling, A., Moraetis, D., Partsinevelos, P., and Oncken, O., 2017. Distinct phases of eustatic and tectonic forcing for late Quaternary landscape evolution southwest Crete, Greece.* Earth Surface Dynamics *5, 1–17, https://doi.org/10.5194/esurf-5-511-2017, 2017.*

*Lin 2011, Three Principles of Soil Change and Pedogenesis in Time and Space. SSSAJ: Volume 75: Number 6.*

*Gallen, S.F., and Wegmann, K.W., 2017, Interactive comment on "Distinct phases of eustatism and tectonics control the Late Quaternary landscape evolution at the southern coastline of Crete" by Vasiliki Mouslopoulou et al.; Clarifying points on response of Mouslopoulou et al. to short comment by Gallen and Wegmann [30 January 2017]: Earth Surface Dynamics, http://www.earth-surf-dynam-discuss.net/esurf-2016-62/esurf-2016-62-SC2-supplement.pdf.*

*Gallen, S.F., and Wegmann, K.W., 2017, Interactive comment on "Distinct phases of eustatism and tectonics control the late Quaternary landscape evolution at the southern coastline of Crete" by Vasiliki Mouslopoulou et al. [8 January 2017]: Earth Surface Dynamics, http://www.earth-surf-dynam-discuss.net/esurf-2016-62/esurf-2016-62-SC1-supplement.pdf.*

**References used in the responses by the authors**

Booth, J.: The response of Mediterranean steepland coastal catchments to base level and climate change, southwestern Crete, Aberystwyth University., 2010.

Gallen, S. F., Wegmann, K. W., Bohnenstiehl, D. R., Pazzaglia, F. J., Brandon, M. T. and Fassoulas, C.: Active simultaneous uplift and margin-normal extension in a forearc high, Crete, Greece, Earth Planet. Sci. Lett., 398, 11–24, doi:10.1016/j.epsl.2014.04.038, 2014.

IUSS Working Group WRB: World Reference Base for Soil Resources 2014, update 2015: International soil classification system for naming soils and creating legends for soil maps., Rome.,

2015.

Mouslopoulou, V., Begg, J., Nicol, A., Oncken, O. and Prior, C.: Formation of Late Quaternary paleoshorelines in Crete, Eastern Mediterranean, Earth Planet. Sci. Lett., 431, 294–307, doi:10.1016/j.epsl.2015.09.007, 2015.

Mouslopoulou, V., Begg, J., Fülling, A., Moraetis, D. and Partsinevelos, P.: Distinct phases of eustatic and tectonic forcing for late Quaternary landscape evolution in southwest Crete , Greece, Earth Surf. Dyn., 5, 511–527, 2017.

Ott, R. F., Gallen, S. F., Wegmann, K. W., Biswas, R. H., Herman, F. and Willett, S. D.: Pleistocene terrace formation, Quaternary rock uplift rates and geodynamics of the Hellenic Subduction Zone revealed from dating of paleoshorelines on Crete, Greece, Earth Planet. Sci. Lett., 525, 115757, doi:10.1016/j.epsl.2019.115757, 2019.

Pope, R., Wilkinson, K., Skourtsos, E., Triantaphyllou, M. and Ferrier, G.: Clarifying stages of alluvial fan evolution along the Sfakian piedmont, southern Crete: New evidence from analysis of post-incisive soils and OSL dating, Geomorphology, 94(1–2), 206–225, doi:10.1016/j.geomorph.2007.05.007, 2008.

Richardson, P. W., Perron, J. T. and Schurr, N. D.: Influences of climate and life on hillslope sediment transport, Geology, 47(5), 423–426, doi:10.1130/G45305.1, 2019.

Scherler, D., Lamb, M. P., Rhodes, E. J. and Avouac, J. P.: Climate-change versus landslide origin of fill terraces in a rapidly eroding bedrock landscape: San Gabriel River, California, Bull. Geol. Soc. Am., 128(7), 1228–1248, doi:10.1130/B31356.1, 2016.

Schwanghart, W., Bernhardt, A., Stolle, A., Hoelzmann, P., Adhikari, B. R., Andermann, C., Tofelde, S., Merchel, S., Rugel, G., Fort, M. and Korup, O.: Repeated catastrophic valley infill following medieval earthquakes in the Nepal Himalaya, Science (80-. )., 351(6269), 147–150, doi:10.1126/science.aac9865, 2016.

Stolle, A., Bernhardt, A., Schwanghart, W., Hoelzmann, P., Adhikari, B. R., Fort, M. and Korup, O.: Catastrophic valley fills record large Himalayan earthquakes, Pokhara, Nepal, Quat. Sci. Rev., 177, 88–103, doi:10.1016/j.quascirev.2017.10.015, 2017.

---

## Author Comment (AC4)

**Line by line responses to Anonymous referee # 1**

This is a very timely contribution when we are slowly moving away from rather simple-minded interpretations of alluvial stratigraphy to take extreme events more into account. That said, my only criticism of the paper is that the theoretical component is not as strong as it should be. Bodies of alluvium that are interpreted to be a result of a change of climate for example may be the sum total of extreme events, the frequency and magnitude of which are modulated by the ambient climate. So, there may not be a substantive difference between the traditional interpretation and what the authors of this paper claim to be stochastic events. I would like to see an **additional paragraph** that sets out the authors' views on this issue.

We thank the reviewer for this constructive feedback. We have added to the discussion section of the revised manuscript (sect. 5.4).

My other comments are more minor, as follows:

1.  Line 22 what is meant by 'intermediate fan'? Clarify.

The term "intermediate fan" refers to its location between the top and bottom alluvial deposits. However, to clarify, we renamed the fan in question "lower fan", as has been done already for the radiocarbon dating report. We have tried to clarify this statement and quote from the revised abstract: "We show that the > 20 m thick lower fan unit, previously thought to be late Pleistocene in age, unconformably buries a paleoshoreline uplifted in the first centuries AD, placing the depositional age of this unit firmly into the Late Holocene." (line 22-24)

2.  Lines 62 and following. The absence of reference to the role of land use in the alluvial stratigraphy of the Mediterranean is puzzling. See the early work of Claudio Vita-Finzi for example. Please include some reference to this phenomenon.

While we acknowledge that hominids have directly and indirectly modified alluvial deposits around the Mediterranean for hundreds of thousands of years through fire, forest clearing, agriculture, animal husbandry, etc., such activity is minimal in our study basin. Native forests were cleared from much of Crete for shipbuilding, agriculture, and olive cultivation, however, the location of Klados catchment on the steep, rocky and hard-to-access southern coast of Crete means that this basin likely experienced very little long-term human alteration of the landscape. With the exception of browsing by wild goats, there was no terracing of hillslopes for agriculture, no planting of olive trees or other wide-spread soil disturbance in the catchment that would manifest itself as part of the alluvial record.

We have added the following sentences to the revised manuscript: "Also, human land use and vegetation cover have been shown to influence sediment dynamics and alluviation patterns, and the Eastern Mediterranean has been central to the investigation of the interplay between climate fluctuations, long-term tectonics, and anthropogenic disturbances (Atherden and Hall, 1999; Benito et al., 2015; Dusar et al., 2011; Thorndycraft and Benito, 2006; Vita-Finzi, 1969)." (line 66-70), and "[...] and is surrounded by steep, 2 km high mountains, which has kept human influence minimal." (line 90)

3. Line 80 please explain why this catchment is anomalous

We have revised this sentence for clarification and added a photograph of a neighbouring river outlet for comparison (Fig. 1c). We quote from the revised text: "However, the thick sequence of several > 20 m thick alluvial fan and terrace deposits preserved in the Klados catchment are anomalous compared to nearby catchments with larger drainage areas (i.e., Samaria) that preserve only minor alluvial deposits." (line 86-88)

4. Line 108-109 what is the evidence for this statement?

We have revised this statement and quote from the new version: "The volumes of these deposits are substantially larger compared to alluvial deposits in larger neighboring catchments and therefore require an unusually high sediment supply input." (line 112-113)

5. Line 165 and following. While there is discussion later on about the accuracy of these C-14 dates from bulk organic matter, please provide a brief preparation here for that later discussion.

We extended this section to include a short discussion on our choice of radiocarbon dating, and the reader is referred to the relevant part in the discussion.

We quote from the revised section: "To constrain the timing of aggradation and incision of the deposits, we radiocarbon-dated bulk organic matter collected from six fine-grained lenses within the deposits. While bulk radiocarbon dating of alluvial sediments will result in larger uncertainties, in this case, it is the only available geochronometric technique given the mineralogy of the sediments and lack of macro-organic material for traditional AMS radiocarbon dating. Additionally, despite uncertainties associated with bulk radiocarbon dating, it is appropriate for discriminating whether or not the sediments are late Pleistocene or Holocene, one of the hypotheses tested with this study. We decided against using luminescence dating because of the sparsity of quartz and feldspar in the local carbonate bedrock and the turbulent mode and the short transport distance that likely result in incomplete bleaching, especially of feldspar grains (Rhodes, 2011). A detailed discussion of uncertainties associated with this method is provided in section 5.1." (line 195-204)

6. Line 228 (and 253) I am unconvinced that these deposits are from sheet flows. I would not expect the shear stresses needed to move the gravel particles can be achieved by sheet flow. Please provide evidence of your claim or perhaps suggest that the deposits are a result of flow in shallow channels.

We agree with the reviewer and change the terminology accordingly. We quote from the revised text: "The upper portions of the alluvial fill units are always layered and fluvially reworked, resembling the planar beds typical of flow in shallow channels (Fig. 4d, e; Blair and McPherson, 2015)" (line 303-305)

7. Line 322 reference here to slackwater deposits may be inappropriate. This term is now used for paleoflood deposits. I suggest that you find an alternative or, if they really are slackwater deposits, please provide more information.

Indeed, slackwater deposits consist of sand and silt, which are deposited when flow velocities are locally reduced during large flood events (Saynor and Erskine, 1993). Descriptions in literature include tributary mouths, widening channels and locations of bedrock or talus obstructions, and overbank deposits on high river terraces (Kochel and Baker, 1988; Pickup et al., 1988; Saynor and Erskine, 1993). In our field area, the deposit in question lies at a tributary mouth, whose outflow was obstructed by one of the valley infills. Consequently, the use of slackwater deposit appears to fit the situation. However, due to this ambiguity, we refrain from categorizing the deposit as slackwater deposit but call them with the more descriptive term of "tributary deposit".

8. Lines 346 and 347. The negative exponents need to be changed.

We thank the reviewer for this remark and have revised the exponents.

9. Line 377 here and elsewhere you refer to immature soil development but I cannot find an argument for their immaturity. This needs to be rectified.

Based on sedimentological investigation, topographic surveys, soil redness indices, and chronometric dating, Pope et al. (2008) interpret the sediment in the Sfakia piedmont 25 km to the east of Klados as deposited during cold stages of the major glacial cycles. In close comparison with photographs of these sites, and a preliminary soil classification during field work, we find that the soils in the Klados catchment are immature throughout the mapping area (IUSS Working Group WRB, 2015). The main evidence comes from soil redness, depth, density, and the extent of the vegetation cover, as we state in section 4.1. We quote from the revised section: "Soils are weakly developed on all three alluvial fill units as is derived from soil redness, depth, density, and vegetation cover (Fig. S5). Moreover, there are no discernable secondary carbonates or other mineral diagnostic horizons related to migration processes, and clay formation is insignificant. The terraces lack fluvic properties and are well-drained, which is why the best categorisation appears to be a calcaric, skeletic Regosol (IUSS Working Group WRB, 2015)." (line 305-309)

To further illustrate this point we added a new supplemental figure S5:

[Figure]

**Figure S5**: Minor soil development on $T_3$ (a), $T_2$ (b), and $T_1$ (c) results in low soil maturity. Typically, a surface horizon of non-degraded organic matter such as pine needles overlies the original alluvial deposits. Soil formation may be accelerated in close proximity to larger plants such as pine trees, but we find no sign of wide-spread pedogenesis. (d) Outcrop "Alta Paleohora" (20 km W of Klados, exact location noted) showing dated MIS 4 alluvial fan material over MIS 5.1 beach deposits (Pope et al., 2008). (e) Outcrop in Paleohora (exact location noted), carbonaceous terrace of MIS 2. B = top soil, C = source rock, K= secondary carbonates, T = clay-enriched (IUSS Working Group WRB, 2015).

10. Line 503 you claim that this catchment is unique but do not explain why. Also see my comment #3 above.

We have modified the section to improve clarity. We quote: "The alluvial deposits in the Klados catchment are volumetrically oversized and immature in soil development compared to other catchments in southern Crete. We have demonstrated that the deposits preserved in the valley are Holocene in age and that following a massive landslide event, the catchment dynamics are best described by rapid and dramatic alternations between valley-wide aggradation and incision. These findings show that the emplacement of the landslide deposit altered catchment dynamics, making Klados more sensitive to external forcing. This change in sensitivity to external forcing makes the Klados fans distinct among the well-studied Pleistocene fans in Crete." (line 596-602)

11. Line 507 please explain why the landslide deposit made this catchment ultra-sensitive to external forcing.

We refer the reader to section 5.4. in our revised manuscript, where we discuss the ultrasensitivity in terms of sediment and water discharge rates. We quote from this revision: "While in each case sediment transport events are likely associated with high-intensity rainstorms, as indicated by the high-energy depositional environments inferred from fan stratigraphy in Klados and Pleistocene fans elsewhere on Crete, the threshold magnitude for a sediment-generating event, whether a rainstorm or seismically-driven ground shaking, in Klados is likely much smaller relative to those that produced the Pleistocene fans. This difference in sensitivity to external forcing makes the Klados fans unique in the context of Pleistocene fans of Crete" (line 602-607)

12. Line 547 this is not a recurrence interval but a frequency. Please change.

This is a good point by the reviewer, which we changed in the revised manuscript.

**References used by the authors in the responses**

Atherden, M. A. and Hall, J. A.: Human impact on vegetation in the White Mountains of Crete since AD 500, The Holocene, 9(2), 183–193, doi:10.1191/095968399673523574, 1999.

Benito, G., Macklin, M. G., Zielhofer, C., Jones, A. F. and Machado, M. J.: Holocene flooding and climate change in the Mediterranean, Catena, 130, 13–33, doi:10.1016/j.catena.2014.11.014, 2015.

Blair, T. C. and McPherson, J. G.: Processes and Forms of Alluvial Fans - Geomorphology of Desert Environments, in Geomorphology of Desert Environments, edited by A. J. Parsons and A. D. Abrahams, pp. 413–467, Springer Science & Business Media., 2015.

Dusar, B., Verstraeten, G., Notebaert, B. and Bakker, J.: Holocene environmental change and its impact on sediment dynamics in the eastern Mediterranean, Earth-Science Rev., 108(3–4), 137–157, doi:10.1016/j.earscirev.2011.06.006, 2011.

IUSS Working Group WRB: World Reference Base for Soil Resources 2014, update 2015: International soil classification system for naming soils and creating legends for soil maps., Rome., 2015.

Kochel, R. C. and Baker, V. R.: Paleoflood analysis using slackwater deposits, in Flood Geomorphology, edited by V. R. Baker, R. C. Kochel, and P. C. Patton, pp. 357–376, Wiley and Sons, New York., 1988.

Pickup, G., Allan, G. and Baker, V. R.: History, palaeochannels and palaeofloods of the Finke river, central Australia, in Fluvial Geomorphology of Australia, edited by R. F. Warner, pp. 177–200, Academic Press, Sidney., 1988.

Pope, R., Wilkinson, K., Skourtsos, E., Triantaphyllou, M. and Ferrier, G.: Clarifying stages of alluvial fan evolution along the Sfakian piedmont, southern Crete: New evidence from analysis of post-incisive soils and OSL dating, Geomorphology, 94(1–2), 206–225, doi:10.1016/j.geomorph.2007.05.007, 2008.

Rhodes, E. J.: Optically Stimulated Luminescence Dating of Sediments over the Past 200,000 Years, Annu. Rev. Earth Planet. Sci., 39(1), 461–488, doi:10.1146/annurev-earth-040610-

133425, 2011.

Saynor, M. J. and Erskine, W. D.: Characteristics and implications of high-level slackwater deposits in the fairlight gorge, nepean river, australia, Mar. Freshw. Res., 44(5), 735–747, doi:10.1071/MF9930735, 1993.

Thorndycraft, V. R. and Benito, G.: Late Holocene fluvial chronology of Spain: The role of climatic variability and human impact, Catena, 66(1–2), 34–41, doi:10.1016/j.catena.2005.07.007, 2006.

Vita-Finzi, C.: The Mediterranean valleys: geological changes in historical times, Cambridge University Press, Cambridge., 1969.